# Optimal Transport-based Conformal Prediction

**Gauthier Thurin** [1]   **Kimia Nadjahi** [1]   **Claire Boyer** [2]

## Abstract

Conformal Prediction (CP) is a principled framework for quantifying uncertainty in black-box learning models, by constructing prediction sets with finite-sample coverage guarantees. Traditional approaches rely on scalar nonconformity scores, which fail to fully exploit the geometric structure of multivariate outputs, such as in multi-output regression or multiclass classification. Recent methods addressing this limitation impose predefined convex shapes for the prediction sets, potentially misaligning with the intrinsic data geometry. We introduce a novel CP procedure handling multivariate score functions through the lens of optimal transport. Specifically, we leverage Monge-Kantorovich vector ranks and quantiles to construct prediction region with flexible, potentially non-convex shapes, better suited to the complex uncertainty patterns encountered in multivariate learning tasks. We prove that our approach ensures finite-sample, distribution-free coverage properties, similar to typical CP methods. We then adapt our method for multi-output regression and multiclass classification, and also propose simple adjustments to generate adaptive prediction regions with asymptotic conditional coverage guarantees. Finally, we evaluate our method on practical regression and classification problems, illustrating its advantages in terms of (conditional) coverage and efficiency.

## 1. Introduction

In various domains, including high-stakes applications, state-of-the-art performances are often achieved by black-box machine learning models. As a result, accurately quantifying the uncertainty of their predictions has become a crit-ical priority. Conformal Prediction (CP, Vovk et al., 2005) has emerged as a compelling framework to address this need, by generating prediction sets with *coverage guarantees* (ensuring they contain the true outcome with a specified confidence level) regardless of the model or data distribution. Most CP methods are thus model-agnostic and distribution-free while being easy to implement, which explains their growing popularity in recent years.

The main idea of CP is to convert a set of *non-conformity scores* into reliable uncertainty sets using *quantiles*. Non-conformity scores are empirical measurements of how unusual a prediction is. For example, in regression, the score can be defined as $|\hat{y} - y|$, where $\hat{y} \in \mathbb{R}$ is the model's prediction and $y \in \mathbb{R}$ the true response (Lei et al., 2018). These scores are central to the CP framework as they encapsulate the uncertainty stemming from both the model and the data, directly influencing the size and shape of the resulting prediction sets. Therefore, the quality of the prediction sets hinges on the relevance of the chosen non-conformity score: while a poorly designed score may still achieve the required coverage guarantee, it often leads to overly conservative or inefficient prediction sets, failing to capture the complex patterns of the underlying data distribution (Angelopoulos & Bates, 2023).

Most CP approaches rely on *scalar* non-conformity scores (*e.g.*, Angelopoulos & Bates, 2023; Romano et al., 2020b; Cauchois et al., 2021; Sesia & Romano, 2021; Lei et al., 2018). Although conceptually simple, such one-dimensional representations can be too restrictive or poorly suited in applications that require multivariate prediction sets. To circumvent this, recently-proposed CP methods seek to incorporate correlations despite the use of scalar scores, by leveraging techniques such as copulas (Messoudi et al., 2021) or ellipsoids (Johnstone & Cox, 2021; Messoudi et al., 2022; Henderson et al., 2024). Nevertheless, these approaches either lack finite-sample coverage guarantees or impose restrictive modeling assumptions that prescribe the shape of the prediction region. Feldman et al. (2023) recently proposed a CP method able to construct more adaptive prediction regions with non-convex shapes, establishing a connection with multivariate quantiles. However, their method (as conformalized quantile regression, Romano et al., 2019) requires intervening in the way the predictor is trained and, therefore, cannot be directly applied

[1]CNRS, Ecole Normale Supérieure, Paris, France [2]Laboratoire de Mathématiques d'Orsay (LMO), Université Paris Saclay, France, and Institut universitaire de France. Correspondence to: Gauthier Thurin <gthurin@mail.di.ens.fr>.

*Proceedings of the $42^{nd}$ International Conference on Machine Learning*, Vancouver, Canada. PMLR 267, 2025. Copyright 2025 by the author(s).

to a black-box model.

**Contributions.** In this work, we introduce a novel general CP framework that accommodates multivariate scores, enabling more expressive representations of prediction errors. The core idea is to leverage *Monge-Kantorovich (MK) quantiles* (Chernozhukov et al., 2017; Hallin et al., 2021), a multivariate extension of traditional scalar quantiles rooted in optimal transport theory with various applications (see *e.g.*, Carlier et al., 2016; Hallin, 2022; Rosenberg et al., 2023). MK quantiles are constructed by mapping multidimensional scores onto a reference distribution. The resulting CP framework, called OT-CP for Optimal Transport-based Conformal Prediction, effectively captures the structure and dependencies within multivariate data while ensuring distribution-free ranks, thanks to the distinctive properties of MK quantiles (Deb & Sen, 2023; Hallin et al., 2021). We demonstrate that OT-CP constructs prediction regions with finite-sample coverage guarantees. These hold for any choice of multivariate score function, which makes OT-CP a robust and practical tool to address complex uncertainty quantification task.

After presenting the general OT-CP methodology with its theoretical guarantees (Section 2), we apply it to two typical learning tasks: multi-output regression (Section 3) and classification (Section 4). For each of these, we use multivariate score functions which, when integrated in OT-CP, yield prediction regions that effectively capture correlations between the score dimensions. In the context of regression, we also develop an extension of OT-CP that conditionally adapts to input covariates, further enhancing the flexibility of our method. Moreover, we show that this adaptive version provably reaches asymptotic conditional coverage. These two case studies serve a dual purpose: they highlight the versatility and user-friendliness of OT-CP while offering concrete frameworks to evaluate its benefits over existing methods through numerical experiments. In doing so, we believe this lays a solid foundation for future explorations of OT-CP across a wider range of applications. The code used to produce the results in this paper can be accessed at this GitHub repository.

## 2. Methodology

### 2.1. Setting

We consider a pre-trained black-box model $\hat{f} : \mathcal{X} \to \mathcal{Y}$, where $\mathcal{X}$ and $\mathcal{Y}$ respectively denote the input and output spaces of the learning task. Assume we have access to a set of $n$ exchangeable observations $(X_i, Y_i) \in \mathcal{X} \times \mathcal{Y}$, not used during the training of $\hat{f}$ and referred to as the *calibration set*. Consider a *score function* $s : \mathcal{X} \times \mathcal{Y} \to \mathbb{R}_+^d$ that produces $d \geq 1$ *non-conformity scores*, measuring the discrepancies between the target $Y_i$ and the prediction $\hat{f}(X_i)$.

Considering a multivariate score in the context of CP departs

from typical strategies, which rely on scalar scores. Such multivariate scores can particularly be useful for quantifying uncertainties, as described in the examples below.

**Example 1** (Multi-output regression)**.** *In multi-output regression, both the response $Y$ and prediction $\hat{f}(X)$ take values in $\mathbb{R}^d$. One can consider multivariate scores $s(Y, \hat{f}(X))$ corresponding to component-wise prediction errors (see Section 3), without the need of aggregating them into a single value (e.g., by considering the mean squared error). Figure 1(a) illustrates 2-dimensional scores in a context of bivariate regression.*

**Example 2** (Multiclass classification)**.** *Consider a classification setting with $K \geq 3$ classes and let $\hat{\pi}(x) = \{\hat{\pi}_k(x)\}_{k=1}^K$ be the estimated class probabilities returned by a classifier for some input $x$. Denote by $\bar{y} = \{\mathbb{1}_{k=y}\}_{k=1}^K$ the one-hot encoding of a label $y$. A multivariate score can be formed as the component-wise absolute difference*

$$s(x, y) = |\hat{\pi}(x) - \bar{y}| \in \mathbb{R}_+^K. \tag{1}$$

*This score retains $K$-dimensional predictive information, allowing for the exploration of correlations between its components. For instance, when $K = 3$, consider two inputs $x_1$ and $x_2$ with output probabilities $\hat{\pi}(x_1) = (0.6, 0.4, 0)$ and $\hat{\pi}(x_2) = (0, 0.4, 0.6)$. For both predictions, assessing the conformity of $y = 2$ with a typical score $1 - \hat{\pi}_y(x)$ used in CP for classification would return the same value of $0.6$. This potentially ignores that co-occurrences between labels $1$ and $2$ might be more frequent than between $2$ and $3$. In contrast, the multivariate alternative (1) distinguishes these two probability profiles, as $s(x_1, y) \neq s(x_2, y)$. This can be more helpful to capture the underlying confusion patterns of the predictor across different label modalities.*

In the rest of the paper, we denote by $\{S_i\}_{i=1}^n = \{s(X_i, Y_i)\}_{i=1}^n$ the scores computed on the calibration set.

### 2.2. Optimal transport toolbox

In the context of conformal prediction, dealing with multivariate scores implies defining an adequate notion of multivariate quantiles. To do so, we view the non-conformity scores $\{S_i\}_{i=1}^n$ through the empirical distribution $\hat{\nu}_n = \frac{1}{n} \sum_{i=1}^n \delta_{S_i}$ and leverage optimal transport (OT) tools, more specifically, *Monge-Kantorovich quantiles*.

**Definition 2.1** (Empirical Monge-Kantorovich ranks, Chernozhukov et al. (2017); Hallin et al. (2021))**.** *Consider the reference rank vectors $\{U_i\}_{i=1}^n$ given by*

$$\forall i \in \{1, \ldots, n\}, \quad U_i = \frac{i}{n}\theta_i, \tag{2}$$

*where $\theta_i$ are i.i.d. random vectors drawn uniformly on the Euclidean sphere $\mathbb{S}^{d-1} = \{\theta \in \mathbb{R}^d : \|\theta\| = 1\}$. The Monge-Kantorovich rank map is defined for any score $s \in \mathbb{R}^d$ as*

$$\mathbf{R}_n(s) = \underset{U_i : 1 \leq i \leq n}{\mathrm{argmax}} \left\{ \langle U_i, s \rangle - \psi_n(U_i) \right\}, \tag{3}$$

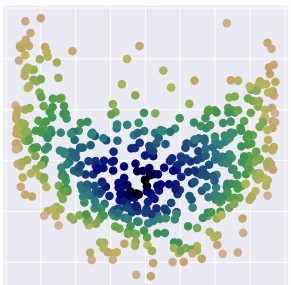 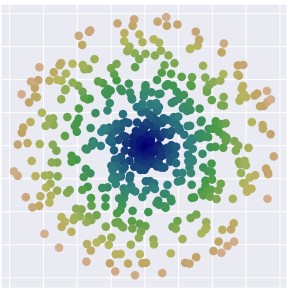

(a) Multivariate scores $\{S_i\}_{i=1}^n$ (b) Reference rank vectors $\{U_i\}_{i=1}^n$

*Figure 1.* Ranking multivariate scores using optimal transport. The colormap encodes how the 2-dimensional scores $\{S_i\}_{i=1}^n$ in (a) are transported onto the reference rank vectors $\{U_i\}_{i=1}^n$ in (b).

*with $\psi_n$ the potential solving the dual of Kantorovich's OT problem, i.e.,*

$$\psi_n = \underset{\varphi}{\operatorname{argmin}} \frac{1}{n} \sum_{i=1}^n \varphi(U_i) + \frac{1}{n} \sum_{i=1}^n \varphi^*(S_i),$$

*where the optimization is performed over the set of lower-semicontinuous convex functions, and $\varphi^*(x) = \sup_u\{\langle x, u \rangle - \varphi(u)\}$ is the Legendre transform of a convex function $\varphi$.*

Note that $\mathbf{R}_n$ verifies $\mathbf{R}_n(S_i) = U_{\sigma_n(i)}$ for $i \in \{1, \ldots, n\}$, where $\sigma_n$ is the solution of the assignment problem

$$\sigma_n = \underset{\sigma \in P_n}{\operatorname{argmin}} \sum_{i=1}^n \|S_i - U_{\sigma(i)}\|^2, \qquad (4)$$

for $P_n$ the set of all permutations of $\{1, \cdots, n\}$.

This transport-based rank map echoes the one-dimensional case, with $\{U_i\}_{i=1}^n$ replacing traditional ranks $\{1, 2, \ldots, n\}$ based on univariate quantile levels $\{\frac{1}{n}, \frac{2}{n}, \ldots, 1\}$. By definition, $\|\mathbf{R}_n(s)\| \in \{\frac{1}{n}, \frac{2}{n}, \ldots, 1\}$ for any $s \in \mathbb{R}^d$. This allows to introduce a specific ordering of $\mathbb{R}^d$, namely

$$s_1 \leq_{\mathbf{R}_n} s_2 \text{ if, and only if, } \|\mathbf{R}_n(s_1)\| \leq \|\mathbf{R}_n(s_2)\|.$$

This multivariate ordering is illustrated in Figure 1. A main virtue is its ability to capture the shape of the underlying probability distribution. Another notable advantage of Definition 2.1 is that the ranks are *distribution-free*: by construction, $\|\mathbf{R}_n(S_1)\|, \ldots, \|\mathbf{R}_n(S_n)\|$ correspond to a random permutation of $\{\frac{1}{n}, \frac{2}{n}, \cdots, 1\}$, regardless of the distribution of the non-conformity scores $\{S_i\}_{i=1}^n$.

This concept of multivariate rank allows the construction of multivariate quantile regions, which play a central role in our methodology, as we will see in the next section.

**Remark 2.2.** The choice of reference rank vectors $\{U_i\}_{i=1}^n$ in Definition 2.1 is flexible and can be tailored to specific needs, provided that $\{S_i\}_{i=1}^n$ and $\{U_i\}_{i=1}^n$ remain independent (Ghosal & Sen, 2022, Remark 3.11). The convention adopted in Definition 2.1 has the merit to fix the ideas and to be appropriate for regression tasks.

### 2.3. OT-based conformal prediction (OT-CP)

We present a new methodology, OT-CP, that leverages optimal transport to perform (split) conformal prediction with multivariate scores. Unlike traditional CP approaches, our method relies on a multivariate perspective to quantify uncertainties and construct prediction regions through Monge-Kantorovich vector quantiles. Given a confidence level $\alpha \in [0, 1]$, the proposed framework consists of three steps:

1. **Multivariate score computation:** Compute the multivariate scores $\{S_i\}_{i=1}^n = \{s(X_i, Y_i)\}_{i=1}^n$ on the calibration set $\{(X_i, Y_i)\}_{i=1}^n$,

2. **Quantile region construction:**

   (a) Split the calibration set into $\mathcal{D}_1$ and $\mathcal{D}_2$ of respective sizes $n_1$ and $n_2$ such that $n_1 + n_2 = n$,

   (b) Compute the MK rank map $\mathbf{R}_{n_1}$ based on $\mathcal{D}_1$,

   (c) Construct the MK quantile region,

   $$\widehat{\mathcal{Q}}_n(\alpha) = \left\{ s : \|\mathbf{R}_{n_1}(s)\| \leq \rho_{\lceil (n_2+1)\alpha \rceil} \right\}, \quad (5)$$

   where $\rho_{\lceil (n_2+1)\alpha \rceil}$ is the $\lceil (n_2 + 1)\alpha \rceil$-th smallest element of $\{\|\mathbf{R}_{n_1}(s(X_i, Y_i))\| : (X_i, Y_i) \in \mathcal{D}_2\}$.

3. **Prediction set computation:** For a test input $X_{\text{test}}$ from the same distribution as the calibration set, return the prediction region,

   $$\widehat{\mathcal{C}}_\alpha(X_{\text{test}}) = \left\{ y \in \mathcal{Y} : s(X_{\text{test}}, y) \in \widehat{\mathcal{Q}}_n(\alpha) \right\}.$$

The key novelty of OT-CP lies in the use of multivariate scores (step 1) along with the construction of an OT-based confidence region (step 2). By definition, this region leverages multivariate quantiles of the empirical distribution of calibration scores, accounting for marginal correlations within their components. Our methodology enables the construction of confidence regions without predefined shapes, thereby aligning better with the underlying data distribution. In step 3, the prediction set for a new input $X_{\text{test}}$ is evaluated through the preimage by the score function of the quantile region. This generalizes the one-dimensional case, where classical quantiles are used to construct prediction sets in the form of intervals.

Extending conformal prediction to the multivariate setting using optimal transport quantiles poses non-trivial challenges, as standard distribution-free arguments do not directly apply. To address these subtleties, we introduce a careful splitting strategy (Kuchibhotla, 2020, §3.3.1) in step 2, which helps to provide explicit theoretical coverage guarantees for our OT-CP method.

**Remark 2.3** (Computational aspects). Our approach requires solving an optimal transport problem between two discrete distributions, each consisting of $n$ points. In the case of univariate scores, this OT problem simplifies to a sorting operation, thus one recovers the standard $O(n \log n)$ cost. In the multivariate case, the OT problem is generally solved via linear programming, inducing a computational complexity of $O(n^3)$. For the numerical experiments carried out in this paper, the computational times remain reasonable, as reported in Appendix D.4. In large-scale settings, efficient approximation methods can reduce the computational complexity to $O(n^2)$ (Peyré et al., 2019), as exploited in the entropic OT-CP methodology of Klein et al. (2025).

### 2.4. Coverage guarantees

Next, we show that the prediction regions constructed with OT-CP are valid, meaning they satisfy the coverage property.

**Theorem 2.4** (Coverage guarantee). *Suppose $\{(X_i, Y_i)\}_{i=1}^n \cup \{(X_{\text{test}}, Y_{\text{test}})\}$ are exchangeable. Let $\alpha \in (0,1)$ such that $\lceil \alpha(n_2 + 1) \rceil \leq n_2$. The prediction region $\widehat{\mathcal{C}}_\alpha$ constructed on $\{(X_i, Y_i)\}_{i=1}^n$ satisfies*

$$\alpha \leq \mathbb{P}\big(Y_{\text{test}} \in \widehat{\mathcal{C}}_\alpha(X_{\text{test}})\big) \leq \alpha + \frac{n_{\text{ties}}}{n_2 + 1}, \quad (6)$$

*where the probability is taken over the joint distribution of $\{(X_i, Y_i)\}_{i=1}^n \cup \{(X_{\text{test}}, Y_{\text{test}})\}$ and $n_{\text{ties}} \in \{1, \ldots, n_2 + 1\}$ is the maximum number of ties in $\{\|\mathbf{R}_{n_1}(s(X_i, Y_i))\| : (X_i, Y_i) \in \mathcal{D}_2\} \cup \{\|\mathbf{R}_{n_1}(S_{\text{test}})\|\}$ (i.e., each distinct value in this sample appears at most $n_{\text{ties}}$ times).*

We present two proof strategies for Theorem 2.4 in Appendices B.1 and B.2. The main challenge lies in extending the desirable properties of univariate quantiles, namely distribution-freeness (see, *e.g.*, Hallin et al., 2021) and stability, to the multivariate setting. However, the stability arguments invoked in standard proofs of the quantile lemma (see, *e.g.*, Tibshirani et al., 2019) do not directly apply here, as they rely on unresolved theoretical questions in optimal transport. To address this difficulty, inspired by Kuchibhotla (2020), we adopt a splitting strategy that enables us to derive a multivariate analogue of the quantile lemma. Indeed, the rank of a new sample score among the $n_2$ calibration scores $\mathcal{D}_2$ is guaranteed to follow a uniform distribution, ensuring valid prediction regions without distributional assumptions.

Theorem 2.4 ensures that, for a given coverage level $\alpha \in (0,1)$, the true label $Y_{\text{test}}$ belongs to the OT-based prediction region $\widehat{\mathcal{C}}_\alpha(X_{\text{test}})$ with probability at least $\alpha$. Moreover, this coverage probability is shown to be of the order of $\alpha$, being upper-bounded by $\alpha + n_{\text{ties}}/(n_2 + 1)$ with $n_2$ the size of the subset $\mathcal{D}_2$ and $n_{\text{ties}} \in \{1, \ldots, n_2 + 1\}$. The factor in this upper bound naturally arises from the discrete feature of the MK rank map (3) and our splitting procedure, which may introduce ties in the ranking process. Note that a tie-breaking rule can be applied if ties occur, as usually done in CP (Angelopoulos & Bates, 2023) to enforce $n_{\text{ties}} = 1$.

While OT-CP can be applied to any model and score function, the next sections focus on specific settings to clearly demonstrate its benefits and potential.

## 3. Multi-Output Regression

This section examines the application of OT-CP for multi-output regression. First, we demonstrate how this approach accommodates arbitrary score distributions, enabling the creation of diverse and data-tailored prediction region shapes. Next, we introduce an extension of our method, called OT-CP+, which incorporates conditional adaptivity to input covariates. We demonstrate its effectiveness both empirically and theoretically, establishing an asymptotic coverage guarantee for OT-CP+.

### 3.1. OT-CP can output non-convex prediction sets

For any feature vector $X \in \mathbb{R}^p$ and response vector $Y \in \mathbb{R}^d$, we aim to conformalize the prediction $\hat{f}(X)$ returned by a given black-box regressor.

**CP methods for multi-output regression.** To further motivate OT-CP in this context, we first review existing conformal strategies, that can be reinterpreted as leaning on multivariate quantile regions. A more comprehensive discussion can be found in Appendix C. One could consider vanilla CP relying on a univariate aggregated score, $s(x, y) = \|y - \hat{f}(x)\|$. This yields spherical prediction regions $\{\hat{f}(x)\} + \text{Ball}_{\|\cdot\|}(\tau_\alpha)$[1] where $\text{Ball}_{\|\cdot\|}(\tau_\alpha)$ is the Euclidean ball of radius $\tau_\alpha > 0$. One can also treat the $d$ components of $Y \in \mathbb{R}^d$ separately to produce prediction regions based on hyperrectangles, $\{\hat{f}(x)\} + \prod_{i=1}^d [a_i, b_i]$ (Neeven & Smirnov, 2018). However, these approaches are often ill-suited to accurately capture the geometry of multivariate distributions. In particular, the output prediction sets (whether spherical or hyperrectangles) can be too large when handling anisotropic uncertainty that varies across different output dimensions. To mitigate this, prior works have introduced scores that account for anisotropy and correlations among the residual dimensions, as with ellipsoidal prediction sets (Messoudi et al., 2020; Johnstone & Cox, 2021; Henderson et al., 2024). Still, this implicitly assumes an

---

[1]The expression involves the Minkowski sum between two sets: for two sets $A$ and $B$, $A + B = \{a + b, \, a \in A, \, b \in B\}$.

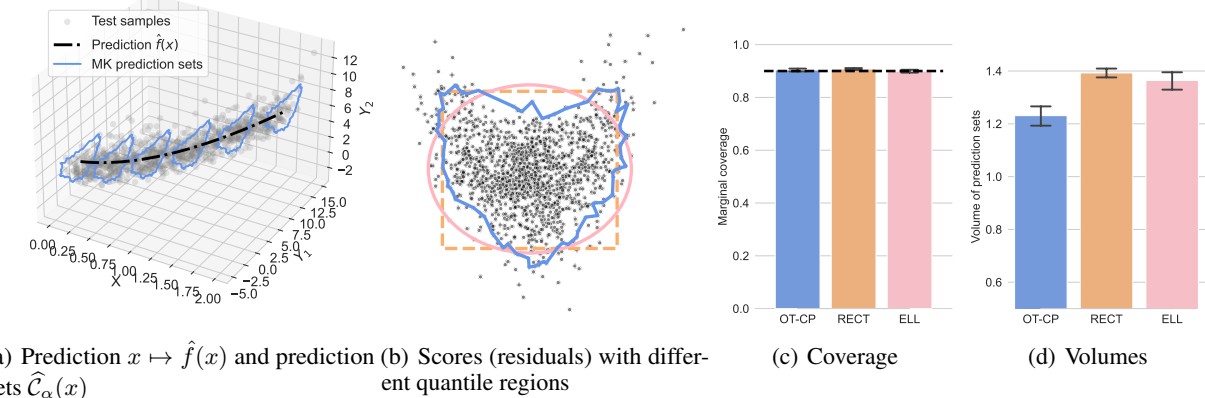

(a) Prediction $x \mapsto \hat{f}(x)$ and prediction sets $\widehat{\mathcal{C}}_\alpha(x)$     (b) Scores (residuals) with different quantile regions     (c) Coverage     (d) Volumes

*Figure 2.* Conformal multi-output regression with OT-CP on simulated data

elliptical distribution for the non-conformity score, thereby compromising the distribution-free nature of the method.

**OT-CP for multi-output regression.** Our strategy consists in applying OT-CP with a multivariate residual as the score,

$$s(x, y) = y - \hat{f}(x) \in \mathbb{R}^d, \tag{7}$$

and yields the following prediction regions

$$\forall x \in \mathbb{R}^p, \quad \widehat{\mathcal{C}}_\alpha(x) = \{\hat{f}(x)\} + \widehat{\mathcal{Q}}_n(\alpha). \tag{8}$$

These sets can take on flexible, arbitrary shapes, that adapt to the calibration error distribution and the underlying data geometry. This key advantage is illustrated concretely in our numerical experiments below.

**Numerical experiments.** In what follows, we study a practical regression problem and compare several CP methods described above: OT-CP for forming prediction regions as in (8), a CP approach producing ellipses (ELL, Johnstone & Cox, 2021), and a simple method creating hyperrectangle (REC, Neeven & Smirnov, 2018), with the miscoverage level adjusted by the Bonferroni correction. We simulate univariate inputs $X \sim \mathrm{Unif}([0, 2])$ with responses $Y \in \mathbb{R}^2$, and we assume that we are given a pre-trained predictor $\hat{f}(x) = (2x^2, (x + 1)^2)$, $x \in \mathbb{R}$. We interpret the score $s(X, Y) = Y - \hat{f}(X)$ as a random vector $\zeta$ distributed from a mixture of Gaussians and independent of $X$, meaning that the distribution of $s(X, Y)$ remains unchanged when conditioned on $X$. Quantile regions for $\alpha = 0.9$ are constructed using $n = 1000$ calibration instances. More implementation details can be found in Appendix D. As expected, OT-CP prediction regions exhibit superior adaptability to the distribution of residuals, whereas hyperrectangles and ellipses tend to be overly conservative (Figures 2(a) and 2(b)). We also compare the methods in terms of empirical coverage on test data (Figure 2(c)) and efficiency (volume of prediction regions, Figure 2(d)). While all approaches adhere to

the $\alpha$-coverage guarantee OT-CP achieves greater efficiency, producing smaller and more precise prediction regions. This highlights that MK quantiles help effectively address uncertainty quantification challenges for multi-output regression.

### 3.2. OT-CP+: an adaptive version

So far, the form of the constructed prediction regions (8) does not depend on the input $X$, as illustrated in Figure 2(a). This uniformity stems from computing quantile regions over the distribution of scores $\{S_i\}_{i=1}^n$ *marginalized* over $\{(X_i, Y_i)\}_{i=1}^n$. In other words, $\{S_i\}_{i=1}^n$ are treated as i.i.d. realizations of $S = Y - \hat{f}(X)$. As a result, while the quantile regions provided by OT-CP effectively capture the global geometry of the scores, they do not adapt to variations in $X$. This lack of adaptivity is inadequate in applications where prediction uncertainties vary between input examples, as discussed by Foygel Barber et al. (2020). The quest for adaptivity led to a rich and diverse literature, see Chernozhukov et al. (2021); Gibbs et al. (2023); Sesia & Romano (2021); Romano et al. (2020a) and references therein, focusing mainly on the design of the univariate score function. We also refer to the concomitant work of Dheur et al. (2025) for a benchmark of CP methods in the case of multi-output regression. In the following, we propose a complementary perspective through conditional MK quantiles on multivariate scores to accommodate input adaptiveness.

**Methodology.** To account for input-dependent uncertainty in the predictions, we introduce OT-CP+, a conformal procedure that computes *adaptive* MK quantile region by leveraging multiple-output quantile regression (del Barrio et al., 2024). Consider the calibration data splitting into $\mathcal{D}_1$ and $\mathcal{D}_2$, as in step 2 of OT-CP. Given a test point $X_{\text{test}}$, we compute the *conditional* MK rank map $\mathbf{R}_k(\cdot | X_{\text{test}})$ based on the $k$-nearest neighbors in $\mathcal{D}_1$ of $X_{\text{test}}$ (and a subsample of $k$ reference vectors). Given a coverage $\alpha \in (0, 1)$,

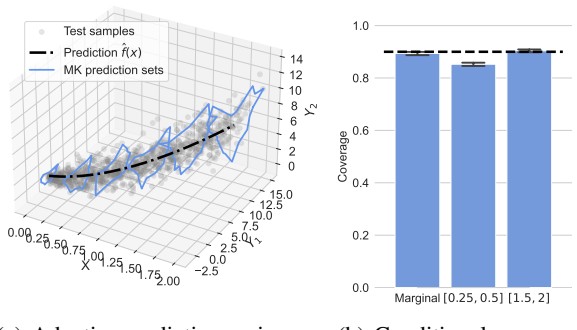

(a) Adaptive prediction regions     (b) Conditional coverage

*Figure 3.* Adaptive conformal regression with OT-CP+

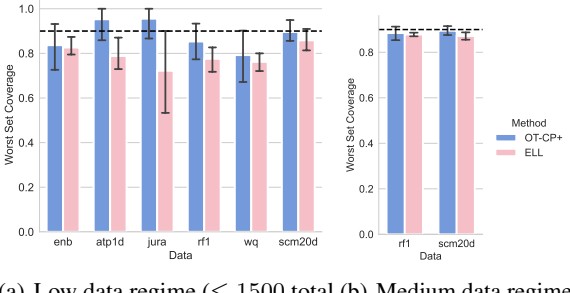

(a) Low data regime ($\leq$ 1500 total samples)   (b) Medium data regime (9000 total samples)

*Figure 4.* Conditional coverage on real datasets of two adaptive conformal procedures for multi-output regression

we calibrate the threshold $\rho_{\lceil (n_2+1)\alpha \rceil}$ using the conditional rank maps $\mathbf{R}_k(\cdot|x)$ for the inputs $x$ associated with $\mathcal{D}_2$. The obtained quantile region is given by,

$$\widehat{\mathcal{Q}}_k(\alpha|X_{\text{test}}) = \left\{ s : \|\mathbf{R}_k(s|X_{\text{test}})\| \leq \rho_{\lceil (n_2+1)\alpha \rceil} \right\}.$$

We defer to Appendix A for a detailed methodology. Hence, OT-CP+ relies on the distribution of $s(X,Y)$ given $X$, which is approximated on a neighborhood of $X$. The prediction regions returned by OT-CP+ are thus given by

$$\widehat{\mathcal{C}}_{\alpha,k}(X_{\text{test}}) = \left\{ \hat{f}(X_{\text{test}}) \right\} + \widehat{\mathcal{Q}}_k(\alpha|X_{\text{test}}). \quad (9)$$

**Experiments on simulated data.** We first consider a similar setting as that of Section 3.1, where the score $s(X,Y)$ is now distributed as $\sqrt{X}\zeta$. Consequently, the variance of the residual increases with $X$, which suggests that wider quantiles should be constructed for larger values of $X$. Figure 3 confirms that OT-CP+ effectively constructs adaptive prediction regions with the desired $\alpha$-coverage. To quantify this more precisely, we evaluate the empirical coverage conditionally on $X$: Figure 3(b) reports box plots of $\mathbb{P}(Y_{\text{test}} \in \widehat{\mathcal{C}}_\alpha(X_{\text{test}})|X_{\text{test}} \in \mathcal{I})$ for several choices of subsets $\mathcal{I} \subset [0,2]$, showing that OT-CP+ satisfies approximate conditional coverage. Note that computational time is larger for OT-CP+ than OT-CP, as it requires to solve multiple OT problems (see Figure 12 in Appendix D.4).

**Experiments on real data.** Next, we evaluate OT-CP+ on real datasets sourced from Mulan (Tsoumakas et al., 2011), with dataset statistics summarized in Table 1. We also implement a concurrent CP method (Messoudi et al., 2022), that is an adaptive extension of the previous ellipsoidal approach (Johnstone & Cox, 2021). Specifically, Messoudi et al. (2022) construct ellipsoidal prediction sets that account for local geometry, by estimating the covariance of $Y|X$ with the $k$-nearest neighbors ($k$NN) of $X$.

We split each dataset into training, calibration, and testing subsets (50%–25%–25% ratio) and train a random forest

model as the regressor. Both methods use a $k$NN step that selects 10% of the calibration set as neighbors for each test point $X_{\text{test}}$. As a coverage metric, we consider the *worst-set coverage*, $\min_{j \in \{1,...,J\}} \mathbb{P}(Y_{\text{test}} \in \widehat{\mathcal{C}}_\alpha(X_{\text{test}})|X_{\text{test}} \in \mathcal{A}_j)$, with $\{\mathcal{A}_j\}_{j \in \{1,...,J\}}$ a partition of the input space tailored to the test data. This metric is conceptually similar to the *worst-slab coverage* (Cauchois et al., 2021), which considers specific partitions in the form of slabs. In our approach, we obtain $J = 5$ regions $\{\mathcal{A}_j\}_{j \in \{1,...,5\}}$ by clustering, *i.e.,* employing (i) a random selection of centroids, and (ii) a $k$NN procedure ensuring that each region contains 10% of the test samples. Empirical results presented in Figure 4 provide evidence supporting the approximate conditional coverage achieved by OT-CP+. Indeed, the worst-set coverage of OT-CP+ remains consistently close to the target level $\alpha = 0.9$ across all datasets, regardless of the sample size or the data dimension. This contrasts with the adaptive ellipsoidal approach, which does not achieve such $\alpha$-coverage and exhibits greater variability. We note, however, that OT-CP+ tend to overcover for datasets with the lowest sample sizes (namely 'atp1d' and 'jura', both with around 350 samples; see Table 1). We also report the marginal coverage, volume and computational time in Figure 13: our results show that OT-CP+ satisfies approximate conditional coverage at the price of larger set sizes on average and runtimes when compared with local ellipsoids.

**Asymptotic conditional coverage.** In the one-dimensional case ($d = 1$), Lei et al. (2018) established the inherent limitation of achieving exact distribution-free conditional coverage in finite samples. However, asymptotic conditional coverage remains attainable under regularity assumptions (Lei et al., 2018; Chernozhukov et al., 2021). OT-CP+ benefits from such a guarantee, leveraging asymptotic properties of quantile regression for MK quantiles (del Barrio et al., 2024). The following assumption is needed.

**Assumption 3.1.** *Suppose that* $(X_1, Y_1), \ldots, (X_n, Y_n)$, $(X_{\text{test}}, Y_{\text{test}})$ *are i.i.d. Assume that for almost every* $x$, *the distribution* $\mathbb{P}_{S|X=x}$ *of* $s(X_{\text{test}}, Y_{\text{test}})$ *given* $X_{\text{test}} = x$

*is Lebesgue-absolutely continuous on its convex support* $\text{Supp}(\mathbb{P}_{S|X=x})$. *For any* $R > 0$, *suppose that its density* $p(\cdot|x)$ *verifies for all* $s \in \text{Supp}(\mathbb{P}_{S|X=x}) \cap \text{Ball}_{\|\cdot\|}(R)$, $\lambda_R^x \leq p(s|x) \leq \Lambda_R^x$.

**Theorem 3.2.** *Let* $k$ *be the number of nearest neighbors used to estimate* $\mathbf{R}_k(\cdot|x)$. *Assume that* $k \to +\infty$ *and* $k/n_1 \to 0$ *as* $n_1 \to +\infty$. *Under Assumption 3.1, the following holds for any* $\alpha \in [0, 1]$, *as* $n_1, n_2 \to +\infty$,

$$\mathbb{P}\big(Y_{\text{test}} \in \widehat{\mathcal{C}}_{\alpha,k}(X_{\text{test}})\big|X_{\text{test}}\big) \xrightarrow{\mathbb{P}} \alpha, \qquad (10)$$

*where* $\widehat{\mathcal{C}}_{\alpha,k}(x) = \{\hat{f}(x)\} + \widehat{\mathcal{Q}}_k(\alpha|x)$ *depends on* $\hat{f}$ *previously learned on (fixed) training data.*

We note that OT-CP+ is not the only way to make OT-CP adaptive: our proposed methodology aims to demonstrate that conditional coverage can be achieved with only a slight modification of the generic OT-CP framework. In addition to being easy to implement, the added $k$-NN step allows us to leverage established results on the consistency of quantiles (Biau & Devroye, 2015; del Barrio et al., 2024), which serve as the foundation for our Theorem 3.2.

# 4. Classification

In this section, we apply OT-CP to multiclass classification. Each data point consists of a feature-label pair $(X, Y) \in \mathbb{R}^p \times \{1, \dots, K\}$, with $K \geq 3$ the number of classes. The given black-box classifier outputs, for any input $X \in \mathbb{R}^p$, a vector $\hat{\pi}(X)$ of estimated class probabilities, where the $k$-th component $\hat{\pi}_k(X)$ is the probability estimate that $X$ belongs to class $k$ (hence, $\sum_{k=1}^K \hat{\pi}_k(X) = 1$).

**CP methods for classification.** Commonly used scores for multiclass classification include the Inverse Probability (IP), $s(x, y) = 1 - \hat{\pi}_y(x)$ and the Margin Score (MS), $s(x, y) = \max_{y' \neq y} \hat{\pi}_{y'}(x) - \hat{\pi}_y(x)$ (Johansson et al., 2017). IP only considers the probability estimate for the correct class label ($\hat{\pi}_y(x)$), whereas MS also involves the most likely incorrect class label ($\max_{y' \neq y} \hat{\pi}_{y'}(x)$). More adaptive options argue in favor of incorporating more class labels in the score function (Romano et al., 2020b; Angelopoulos et al., 2021; Melki et al., 2024). The idea is to rank the labels from highest to lowest confidence (by sorting the probability estimates as $\hat{\pi}_{(y_1)}(x) \geq \hat{\pi}_{(y_2)}(x) \geq \cdots \geq \hat{\pi}_{(y_K)}(x)$), then return the labels such that the total confidence (*i.e.*, the cumulative sum) is at least $\alpha$. It is worth noting that this strategy stems from a notion of *generalized conditional quantile function* (Romano et al., 2020b), by analogy with $\inf_{c \in \mathbb{R}} \{\mathbb{P}(Y \leq c | X = x) \geq \alpha\}$.

**OT-CP for multiclass classification.** As an alternative CP method for this problem, we propose using OT-CP with the following multivariate score,

$$s(X, Y) = |\bar{Y} - \hat{\pi}(X)| \in \mathbb{R}_+^K, \qquad (11)$$

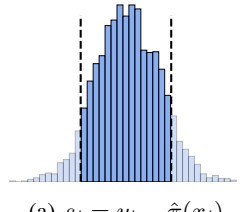 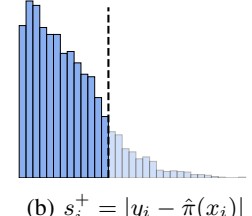

(a) $s_i = y_i - \hat{\pi}(x_i)$  (b) $s_i^+ = |y_i - \hat{\pi}(x_i)|$

*Figure 5.* Ordering must depend on the chosen scores: (a) Center-outward for signed errors, (b) Left-to-right for absolute errors

where the absolute value is taken component-wise and $\bar{Y} = (\mathbb{1}_{Y=k})_{k=1}^K$ denotes the one-hot encoding of $Y$. One can remark in passing that $\|s(x, y)\|_1 = 2(1 - \hat{\pi}_y(x))$, which corresponds to the aforementioned IP scalar score. Our OT-CP procedure builds upon generalized quantiles to take into account *all* the components of $\hat{\pi}_y(x)$ (and not only the largest values) and to capture the correlations between them. The score in (11) takes values in $\mathbb{R}_+^K$ and naturally induces a *left-to-right* ordering. This contrasts with the score function used in our previous application, multi-output regression, where the ordering is center-outward. To further clarify this difference, let us focus on a single component of the score, $s(x, y)_k$, for simplicity. A center-outward interval of the form of $[q_{\alpha/2}, q_{1-\alpha/2}]$ applied to $s(x, y)_k$ excludes lower values from $[0, q_{\alpha/2})$ (Figure 5(a)). This exclusion is problematic for the score structure induced by (11), since lower values of $s(x, y)_k$ indicate greater conformity between $x$ and the ground-truth $y$. In this context, left-to-right ordering is more appropriate, as illustrated in Figure 5(b).

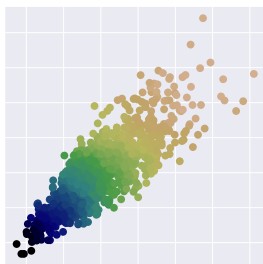 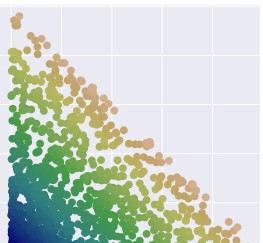

(a) Multivariate scores $\{S_i\}_{i=1}^n$ corresponding to absolute errors in $\mathbb{R}_+^d$

(b) Positive reference rank vectors $\{U_i\}_{i=1}^n$

*Figure 6.* Positive reference ranks for a left-to-right ordering. The colormap encodes how the 2-dimensional scores $\{S_i\}_{i=1}^n$ in (a) are transported onto the reference rank vectors $\{U_i\}_{i=1}^n$ in (b)

A left-to-right ordering can be easily achieved by making a slight adjustment to Definition 2.1: we choose the reference rank vectors as $U_i = \frac{i}{n}\theta_i^+$, where $\theta_i^+$ is uniformly sampled in $\{\theta \in \mathbb{R}_+^d : \|\theta\|_1 = 1\}$. As depicted in Figure 6, the resulting MK ranks reflect the desired left-to-right ordering.

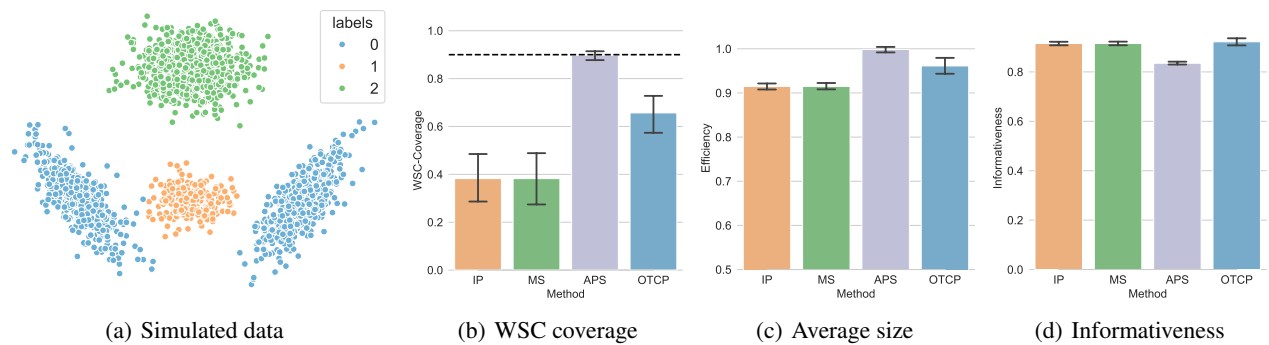

(a) Simulated data      (b) WSC coverage      (c) Average size      (d) Informativeness

*Figure 7.* Conformal classification by Quadratic Discriminant Analysis on simulated data

It is worth noting that this adjustment is fully compatible with the general definition of MK quantiles, which is flexible enough to accommodate arbitrary reference distributions (see Remark 2.2). Based on this choice of score (11) and reference rank vectors, OT-CP generates the following prediction sets,

$$\widehat{\mathcal{C}}_\alpha(x) = \left\{ \bar{y} \in \{0,1\}^K : |\bar{y} - \hat{\pi}(x)| \in \widehat{\mathcal{Q}}_n(\alpha) \right\},$$

where the region $\widehat{\mathcal{Q}}_n(\cdot)$ is constructed from $\{U_i\} = \{\frac{i}{n}\theta_i^+\}$.

**Numerical experiments.** We compare OT-CP against IP, MS and APS scores in terms of worst-case coverage (WSC, measuring conditional coverage, as proposed in Romano et al. (2020b)), efficiency (average size of the predicted set) and informativeness (average number of predicted singletons). More implementation details are given in Appendix D.

We start by simulating data according to a Gaussian mixture model, represented in Figure 7(a) and we consider a pretrained classifier based on Quadratic Discriminant Analysis. Figures 7(b) to 7(d) outline that OT-CP successfully retains the efficiency and informativeness—hallmarks of IP and MS—while simultaneously enhancing conditional coverage on $X$, akin to the improvements achieved by APS. These results highlight that OT-CP effectively handles arbitrary probability profiles by leveraging the entire softmax output, rather than relying solely on its sum, to construct more informative and meaningful prediction sets.

The relevance of OT-CP is also confirmed on real datasets. In Figure 8 and Figure 9, we present the results for a random forest on MNIST and Fashion-MNIST. Additional numerical experiments are provided in Appendix D. Interestingly, despite not being explicitly designed for this purpose, OT-CP achieves conditional coverage with respect to the label on par with APS, where IP and MS fall short, as highlighted in Figure 9. In addition, OT-CP maintains the efficiency and informativeness of IP and MS, offering a convenient balance across all the considered metrics, as one can observe

in Figure 8. We finally emphasize that the numerical experiments were designed as prototypes to demonstrate how OT-CP can be seamlessly and effectively adapted to typical classification tasks. The focus is on demonstrating a useful application of our general framework, which already shows several benefits while remaining conceptually simple.

## 5. Conclusion and perspectives

We have introduced a general and versatile framework for conformal prediction grounded in optimal transport theory. This approach not only revisits classical CP methods based on scalar scores, but also extends easily to handle multivariate scores in a novel and robust manner, thanks to the inherent properties of Monge-Kantorovich quantiles. The OT-CP methodology is flexible, enabling the construction of prediction regions tailored to diverse scenarios, besides being well-suited to capture complex uncertainty structures.

However, in the setting of multi-output regression, our approach may be observed to be more conservative than the ellipsoidal one. We identify opportunities for improvement in this direction by employing more suitable reference distributions for instance. Moreover, the flexibility of the approach, rooted in the Monge-Kantorovich quantile formulation, comes at the cost of increased computational complexity compared to conformal methods based on univariate scores. An alternative transport-based method to achieve better computational and statistical efficiency with respect to the scale of the problem is to use entropic maps, as in the concomitant work by Klein et al. (2025).

In addition, the OT-CP methodology could be adapted to new learning tasks. One might think of multi-label classification where the multivariate score (11) immediately applies by replacing the one-hot encoding by a multi-hot encoding, see, *e.g.,* Katsios & Papadopoulos (2024) for related ellipsoidal inference. One could also explore the development of more sophisticated multivariate scores, potentially building on existing alternatives (*e.g.,* Tumu et al. (2024);

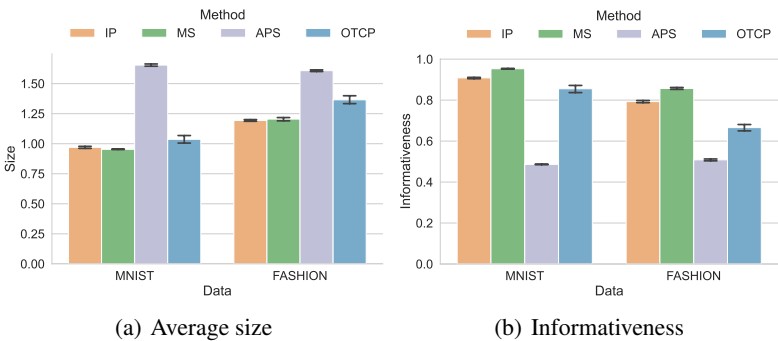

(a) Average size       (b) Informativeness

*Figure 8.* Classification metrics on MNIST and Fashion-MNIST, results averaged over the 10 labels

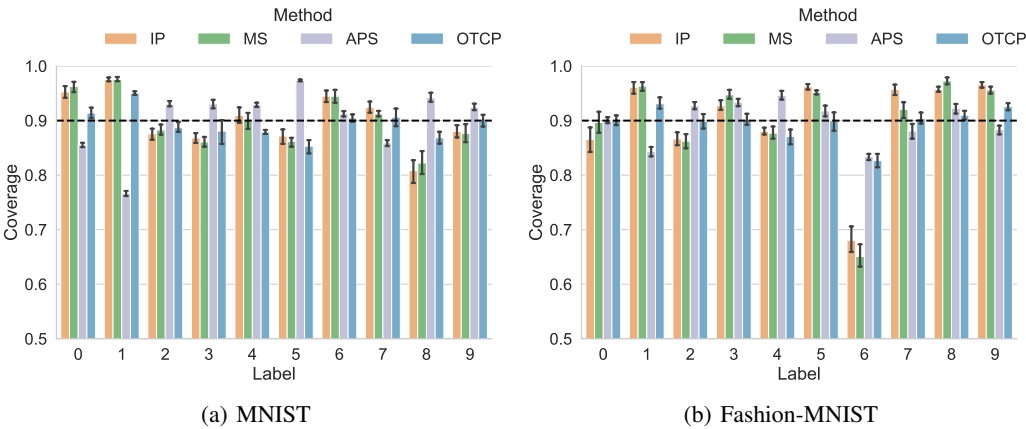

(a) MNIST       (b) Fashion-MNIST

*Figure 9.* Label-wise coverage on $K = 10$ classes of MNIST and Fashion-MNIST

Wang et al. (2023); Plassier et al. (2024) for regression; and Angelopoulos et al. (2021); Melki et al. (2024) for classification). Indeed, our numerical experiments demonstrate that basic multivariate scores can outperform classical univariate counterparts, providing a supplementary motivation for pursuing into this direction.

## Impact Statement

This paper presents work whose goal is to advance the field of Machine Learning. There are many potential societal consequences of our work, none which we feel must be specifically highlighted here.

## Acknowledgments

The authors would like to thank Gilles Blanchard for helpful discussions regarding exchangeability and finite-sample guarantees. This work benefited from state aid managed by the National Research Agency ANR-23-IACL-0008 under France 2030, for the project PR[AI]RIE-PSAI.

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

# A. Detailed methodology for OT-CP+

The OT-CP+ procedure is described in detail below.

1. **Multivariate score computation:** Compute the multivariate scores $\left(s(X_i, Y_i)\right)_{i=1}^n$ on the calibration set $(X_i, Y_i)_{i=1}^n$.

2. **Conditional quantile region construction:**

    (a) Split the calibration set into $\mathcal{D}_1$ and $\mathcal{D}_2$ of respective sizes $n_1$ and $n_2$ such that $n_1 + n_2 = n$. In what follows, for any $x$, we refer to $\mathbf{R}_k(\cdot|X = x)$ as the conditional MK rank map based on the $k$-nearest neighbors of $x$ in $\mathcal{D}_1$ (using $k$ reference vectors).

    (b) For each $(X_i, Y_i) \in \mathcal{D}_2$, compute the conditional MK rank map $\mathbf{R}_k(\cdot|X = X_i)$, then compute $\rho_{\lceil (n_2+1)\alpha \rceil}$ as the $\lceil (n_2 + 1)\alpha \rceil$-th smallest element of,

    $$\left\{ \|\mathbf{R}_k(s(X_i, Y_i)|X = X_i)\| \; : \; (X_i, Y_i) \in \mathcal{D}_2 \right\}.$$

    (c) For any $x$, compute the conditional MK rank map $\mathbf{R}_k(\cdot|X = x)$ and define the conditional MK quantile region as

    $$\widehat{\mathcal{Q}}_k(\alpha|X = x) = \left\{ s : \|\mathbf{R}_k(s|X = x)\| \leq \rho_{\lceil (n_2+1)\alpha \rceil} \right\}. \tag{12}$$

3. **Prediction set computation:** For a test input $X_{\text{test}}$ drawn from the same distribution as the calibration set, return the prediction region

$$\widehat{\mathcal{C}}_{\alpha,k}(X_{\text{test}}) = \left\{ y \in \mathcal{Y} : s(X_{\text{test}}, y) \in \widehat{\mathcal{Q}}_k(\alpha|X = X_{\text{test}}) \right\}.$$

Hence, the main difference from OT-CP lies in using the procedure by del Barrio et al. (2024) to compute conditional MK rank maps $\mathbf{R}_k(\cdot|X = x)$, thereby providing greater adaptivity compared to the standard MK rank map $\mathbf{R}_n(\cdot)$ defined in (3).

# B. Proofs

This section contains the detailed proofs of our theoretical results. For clarity, and without loss of generality, we assume that the calibration set $\{(X_i, Y_i)\}_{i=1}^n$ is split into $\mathcal{D}_1 = \{(X_i, Y_i)\}_{i=1}^{n_1}$ and $\mathcal{D}_2 = \{(X_{n_1+i}, Y_{n_1+i})\}_{i=1}^{n_2}$, where $n_1 + n_2 = n$.

## B.1. Proof of Theorem 2.4 (marginal coverage guarantee)

The proof of Theorem 2.4 consists in extending the reasoning for the traditional quantile lemma (*e.g.,* Lemma 2 in Romano et al. (2019)) to a multivariate setting.

*Proof of Theorem 2.4.* For $k \in \{1, \ldots, n_2\}$, let $S_{(k,n_2)}$ be the $k$-th smallest score among $\{S_{n_1+i}\}_{i=1}^{n_2}$, where $S_{n_1+i} = s(X_{n_1+i}, Y_{n_1+i})$ and the ordering is induced by the transport map $\mathbf{R}_{n_1}$ computed on $\mathcal{D}_1$. Hence,

$$\|\mathbf{R}_{n_1}(S_{(1,n_2)})\| \leq \|\mathbf{R}_{n_1}(S_{(2,n_2)})\| \leq \cdots \leq \|\mathbf{R}_{n_1}(S_{(n_2,n_2)})\|, \tag{13}$$

which, using the definition of $\leq_{\mathbf{R}_{n_1}}$, is equivalently written as

$$S_{(1,n_2)} \leq_{\mathbf{R}_{n_1}} S_{(2,n_2)} \leq_{\mathbf{R}_{n_1}} \cdots \leq_{\mathbf{R}_{n_1}} S_{(n_2,n_2)}. \tag{14}$$

By construction, $\rho_{\lceil \alpha(n_2+1) \rceil} = \|\mathbf{R}_{n_1}(S_{(\lceil \alpha(n_2+1) \rceil, n_2)})\|$, where $\alpha \in (0, 1)$ and $\lceil \alpha(n_2 + 1) \rceil \leq n_2$, ensuring that the order statistic $\rho_{\lceil \alpha(n_2+1) \rceil}$ is well defined. The quantile region $\widehat{\mathcal{Q}}_n(\alpha)$ can be characterized as

$$S_{\text{test}} \in \widehat{\mathcal{Q}}_n(\alpha) \qquad \Longleftrightarrow \qquad S_{\text{test}} \leq_{\mathbf{R}_{n_1}} S_{(\lceil \alpha(n_2+1) \rceil, n_2)}. \tag{15}$$

Denote by $S_{(k,n_2+1)}$ the $k$-th smallest value (with respect to $\leq_{\mathbf{R}_{n_1}}$) within $\{S_{n_1+1}, \ldots, S_n, S_{\text{test}}\}$. Here, we view $S_{\text{test}}$ as inserted into the ordered list $\{S_{(k,n_2)}\}_{k=1}^{n_2}$, yielding a new ordered list of size $n_2 + 1$. Then, $S_{\text{test}} = S_{(i_0+1,n_2+1)}$, with

$$S_{(1,n_2)} \leq_{\mathbf{R}_{n_1}} \cdots \leq_{\mathbf{R}_{n_1}} S_{(i_0,n_2)} \leq_{\mathbf{R}_{n_1}} S_{\text{test}} \leq_{\mathbf{R}_{n_1}} S_{(i_0+1,n_2)} \leq_{\mathbf{R}_{n_1}} \cdots \leq_{\mathbf{R}_{n_1}} S_{(n_2,n_2)}. \tag{16}$$

As a direct consequence of (16), we have

- for $k \leq i_0$, $S_{(k,n_2+1)} = S_{(k,n_2)}$,

- for $k = i_0 + 1$, $S_{(i_0+1,n_2+1)} = S_{\text{test}} \leq_{\mathbf{R}_{n_1}} S_{(i_0+1,n_2)}$,

- for $k > i_0 + 1$, $S_{(k,n_2+1)} = S_{(k-1,n_2)} \leq_{\mathbf{R}_{n_1}} S_{(k,n_2)}$.

Therefore, for any $k \in \{1, \ldots, n_2\}$, $S_{(k,n_2+1)} \leq_{\mathbf{R}_{n_1}} S_{(k,n_2)}$. Hence, if $S_{\text{test}} \leq_{\mathbf{R}_{n_1}} S_{(k,n_2+1)}$, then $S_{\text{test}} \leq_{\mathbf{R}_{n_1}} S_{(k,n_2)}$. We show that the reciprocal also holds. Assume that $S_{\text{test}} \leq_{\mathbf{R}_{n_1}} S_{(k,n_2)}$. By (16), this implies $k \geq i_0 + 1$, and in this case, $S_{(k,n_2+1)}$ is the larger of $S_{\text{test}}$ and $S_{(k-1,n_2)}$ (with respect to $\leq_{\mathbf{R}_{n_1}}$). Putting everything together, we showed that

$$S_{\text{test}} \leq_{\mathbf{R}_{n_1}} S_{(k,n_2)} \qquad \Longleftrightarrow \qquad S_{\text{test}} \leq_{\mathbf{R}_{n_1}} S_{(k,n_2+1)}.$$

Thus, $\mathbb{P}\big(S_{\text{test}} \leq_{\mathbf{R}_{n_1}} S_{(k,n_2)}\big) = \mathbb{P}\big(S_{\text{test}} \leq_{\mathbf{R}_{n_1}} S_{(k,n_2+1)}\big)$. Hence, for any $\alpha \in [0,1]$,

$$\mathbb{P}\big(S_{\text{test}} \in \widehat{\mathcal{Q}}_n(\alpha)\big) = \mathbb{P}\big(S_{\text{test}} \leq_{\mathbf{R}_{n_1}} S_{(\lceil \alpha(n_2+1)\rceil, n_2+1)}\big). \tag{17}$$

Recall that $n_{\text{ties}}$ denotes the maximum number of ties in $\{\mathbf{R}_{n_1}(S_{n_1+1}), \ldots, \mathbf{R}_{n_1}(S_{n_1+n_2}), \mathbf{R}_{n_1}(S_{\text{test}})\}$. By definition of $S_{(k,n_2+1)}$, the proportion of elements from $\{S_1, \cdots, S_n, S_{\text{test}}\}$ that are less than or equal to $S_{(k,n_2+1)}$ (with respect to $\leq_{\mathbf{R}_{n_1}}$) lies between $k/(n_2+1)$ and $(k-1+n_{\text{ties}})/(n_2+1)$, due to the possible presence of ties.

Since $\{(X_i, Y_i)\}_{i=1}^n$ is assumed to be exchangeable, and $\mathbf{R}_{n_1}$ is computed using $\mathcal{D}_1$, then $\{\|\mathbf{R}_{n_1}(S_{n_1+1})\|, \cdots, \|\mathbf{R}_{n_1}(S_{n_1+n_2})\|, \|\mathbf{R}_{n_1}(S_{\text{test}})\|\}$ is exchangeable (see Kuchibhotla, 2020, Proposition 3). Therefore, by (17),

$$\frac{\lceil \alpha(n_2+1)\rceil}{n_2+1} \leq \mathbb{P}\big(S_{\text{test}} \in \widehat{\mathcal{Q}}_n(\alpha)\big) \leq \frac{\lceil \alpha(n_2+1)\rceil - 1 + n_{\text{ties}}}{n_2+1},$$

and we can conclude that

$$\alpha \leq \mathbb{P}\big(S_{\text{test}} \in \widehat{\mathcal{Q}}_n(\alpha)\big) \leq \alpha + \frac{n_{\text{ties}}}{n_2+1}.$$

$\square$

## B.2. Alternative proof of Theorem 2.4

We provide an alternative proof of Theorem 2.4, which is similar in essence to the previous one but based on another perspective. To this end, we recall a variant of the traditional quantile lemma adapted to our needs (*i.e.,* when there are ties in the ranks) and detail its proof for completeness.

**Lemma B.1.** *(Quantile lemma, Lei et al., 2018) Suppose $(U_1, \ldots, U_{n+1})$ is an exchangeable sequence of random variables in $\mathbb{R}$. Then, for any $\beta \in (0,1)$,*

$$\mathbb{P}(U_{n+1} \leq U_{(\lceil \beta(n+1)\rceil)}) \geq \beta \tag{18}$$

*Additionally, assume that the maximum number of ties (i.e., identical values) in $(U_1, \ldots, U_{n+1})$ is $n_{\text{ties}}$. Then,*

$$\mathbb{P}(U_{n+1} \leq U_{(\lceil \beta(n+1)\rceil)}) \leq \beta + \frac{n_{\text{ties}}}{n+1} \tag{19}$$

*The probabilities are taken over the joint distribution of $(U_1, \ldots, U_{n+1})$.*

*Proof.* By exchangeability of $(U_1, \ldots, U_{n+1})$, for any $i \in \{1, \ldots, n+1\}$,

$$\mathbb{P}(U_{n+1} \leq U_{(\lceil \beta(n+1)\rceil)}) = \mathbb{P}(U_i \leq U_{(\lceil \beta(n+1)\rceil)}). \tag{20}$$

Therefore,

$$\mathbb{P}(U_{n+1} \leq U_{(\lceil \beta(n+1)\rceil)}) = \frac{1}{n+1} \sum_{i=1}^{n+1} \mathbb{P}(U_i \leq U_{(\lceil \beta(n+1)\rceil)}) \tag{21}$$

$$= \frac{1}{n+1} \mathbb{E}\left[ \sum_{i=1}^{n+1} \mathbb{1}_{U_i \leq U_{(\lceil \beta(n+1)\rceil)}} \right] \tag{22}$$

$$= \frac{1}{n+1} \mathbb{E}\left[ \sum_{i=1}^{n+1} \mathbb{1}_{U_i < U_{(\lceil \beta(n+1)\rceil)}} + \mathbb{1}_{U_i = U_{(\lceil \beta(n+1)\rceil)}} \right] \tag{23}$$

Since $U_{(\lceil \beta(n+1)\rceil)}$ is the $\lceil \beta(n+1)\rceil$-th smallest value of $(U_1, \ldots, U_{n+1})$, then $\sum_{i=1}^{n+1} \mathbb{1}_{U_i \leq U_{(\lceil \beta(n+1)\rceil)}} \geq \lceil \beta(n+1)\rceil$, and by (22),

$$\mathbb{P}(U_{n+1} \leq U_{(\lceil \beta(n+1)\rceil)}) \geq \frac{1}{n+1}\mathbb{E}\left[\lceil \beta(n+1)\rceil\right]$$
$$\geq \frac{\lceil \beta(n+1)\rceil}{n+1} \geq \beta\,.$$

Additionally, under the assumption that no value appears more than $n_{\text{ties}}$ times among the $n+1$ exchangeable variables, and based on (23), $\sum_{i=1}^{n+1} \mathbb{1}_{U_i \leq U_{(\lceil \beta(n+1)\rceil)}} \leq \lceil \beta(n+1)\rceil - 1 + n_{\text{ties}}$. We can conclude that,

$$\mathbb{P}(U_{n+1} \leq U_{(\lceil \beta(n+1)\rceil)}) \leq \frac{\lceil \beta(n+1)\rceil - 1 + n_{\text{ties}}}{n+1} \leq \beta + \frac{n_{\text{ties}}}{n+1}\,.$$

$\square$

By using Lemma B.1 along with the properties of Monge-Kantorovich rank maps on our split calibration set, we can prove Theorem 2.4 as follows.

*Alternative proof of Theorem 2.4.* By construction of the prediction region, we have

$$\left\{Y_{\text{test}} \in \widehat{\mathcal{C}}_\alpha(X_{\text{test}})\right\} = \left\{S_{\text{test}} \in \widehat{\mathcal{Q}}_n(\alpha)\right\} = \left\{\|\mathbf{R}_{n_1}(S_{\text{test}})\| \leq \rho_{\lceil (n_2+1)\alpha\rceil}\right\}\,.$$

Therefore,

$$\mathbb{P}(Y_{\text{test}} \in \widehat{\mathcal{C}}_\alpha(X_{\text{test}})) = \mathbb{P}\left(\|\mathbf{R}_{n_1}(S_{\text{test}})\| \leq \rho_{\lceil (n_2+1)\alpha\rceil}\right)\,. \tag{24}$$

For $k \in \{1, \ldots, n_2\}$, let $S_{(k,n_2)}$ be the $k$-th smallest score among $\{S_{n_1+i}\}_{i=1}^{n_2}$, where $S_{n_1+i} = s(X_{n_1+i}, Y_{n_1+i})$ with respect to the ordering $\leq_{\mathbf{R}_{n_1}}$ (4). Hence,

$$\|\mathbf{R}_{n_1}(S_{(1,n_2)})\| \leq \|\mathbf{R}_{n_1}(S_{(2,n_2)})\| \leq \cdots \leq \|\mathbf{R}_{n_1}(S_{(n_2,n_2)})\|\,. \tag{25}$$

Additionally, denote by $S_{(k,n_2+1)}$ the $k$-th smallest value (with respect to $\leq_{\mathbf{R}_{n_1}}$) within $\{S_{n_1+1}, \ldots, S_n, S_{\text{test}}\}$. We know that $\|\mathbf{R}_{n_1}(S_{\text{test}})\| \leq \|\mathbf{R}_{n_1}(S_{(k,n_2)})\|$ if, and only if, $\|\mathbf{R}_{n_1}(S_{\text{test}})\| \leq \|\mathbf{R}_{n_1}(S_{(k,n_2+1)})\|$ (*e.g.*, see the proof of Lemma 2 in Romano et al. (2019)).

For $\alpha \in (0,1)$ such that $\lceil \alpha(n_2+1)\rceil \leq n_2$, recall that $\rho_{\lceil \alpha(n_2+1)\rceil}$ is the $\lceil \alpha(n_2+1)\rceil$-th smallest value in $\{\|\mathbf{R}_{n_1}(S_{n_1+i})\|\}_{i=1}^{n_2}$, *i.e.*, $\rho_{\lceil \alpha(n_2+1)\rceil} = \|\mathbf{R}_{n_1}(S_{(\lceil \alpha(n_2+1)\rceil,n_2)})\|$. Therefore, $\|\mathbf{R}_{n_1}(S_{\text{test}})\| \leq \rho_{\lceil \alpha(n_2+1)\rceil}$ if and only if $\|\mathbf{R}_{n_1}(S_{\text{test}})\| \leq \|\mathbf{R}_{n_1}(S_{(\lceil \alpha(n_2+1)\rceil,n_2+1)})\|$. By (24), we deduce that

$$\mathbb{P}(Y_{\text{test}} \in \widehat{\mathcal{C}}_\alpha(X_{\text{test}})) = \mathbb{P}(\|\mathbf{R}_{n_1}(S_{\text{test}})\| \leq \|\mathbf{R}_{n_1}(S_{(\lceil \alpha(n_2+1)\rceil,n_2+1)})\|)\,. \tag{26}$$

As mentioned in Appendix B.1, $(\|\mathbf{R}_{n_1}(S_{n_1+i})\|)_{i=1}^{n_2} \cup \{\|\mathbf{R}_{n_1}(S_{\text{test}})\|\}$ is exchangeable since $\{(X_i, Y_i)\}_{i=1}^{n}$ is assumed to be exchangeable and $\mathbf{R}_{n_1}$ is computed on $\mathcal{D}_1$ and then applied on scores computed on a separated set, namely $\mathcal{D}_2 \cup \{(X_{\text{test}}, Y_{\text{test}})\}$ (see Kuchibhotla, 2020, Proposition 3). We can thus conclude by applying the quantile lemma to $(\|\mathbf{R}_{n_1}(S_{n_1+i})\|)_{i=1}^{n_2} \cup \{\|\mathbf{R}_{n_1}(S_{\text{test}})\|\}$, which is stated and proved in Lemma B.1 for completeness.

$\square$

## B.3. Proof of Theorem 3.2 (asymptotic conditional coverage)

As in Section 3.2, we denote by $\mathbf{R}_k(\cdot|x)$ the conditional empirical MK rank map based on the $k$-nearest neighbors in $\mathcal{D}_1$ of $x$ and by $\widehat{\mathcal{Q}}_k(\alpha|x)$ the conditional MK quantile region of level $\alpha \in [0,1]$ (del Barrio et al., 2024). By assumption, $k$ is a function of $n_1$ satisfying $k \to +\infty$ and $k/n_1 \to 0$ as $n_1 \to +\infty$. For clarity, we omit the explicit dependence of $k$ on $n_1$ in our notation.

By definition of our prediction regions (9), the desired result (10) can also be written as,

$$\mathbb{P}\left(\|\mathbf{R}_k(S_{\text{test}}|X_{\text{test}})\| \leq \rho_{\lceil (n_2+1)\alpha\rceil} \Big| X_{\text{test}}\right) \xrightarrow{\mathbb{P}} \alpha \quad \text{as } n_1, n_2 \to +\infty\,.$$

By applying Corollary 3.4 from del Barrio et al. (2024), we know that

$$\forall \alpha \in [0,1], \quad \mathbb{P}\Big(\|\mathbf{R}_k(S_{\text{test}}|X_{\text{test}})\| \leq \alpha \Big| X_{\text{test}}\Big) \xrightarrow{\mathbb{P}} \alpha \quad \text{as } k, n_1 \to +\infty. \tag{27}$$

Therefore, the main technical challenge of our proof is to understand how the asymptotic convergence guarantee in (27) interacts with our coverage level $\rho_{\lceil (n_2+1)\alpha \rceil}$. To address this, we will rely on the following lemma.

**Lemma B.2.** *For all $\alpha \in [0,1]$ and $n = n_1 + n_2$, $\lim_{n_1, n_2 \to +\infty} \rho_{\lceil (n_2+1)\alpha \rceil} = \alpha$.*

*Proof.* Let $\alpha \in [0,1]$ and for $k \in \mathbb{N}^*$, let $f_k(X_{\text{test}}) = \mathbb{P}\big(\|\mathbf{R}_k(S_{\text{test}}|X_{\text{test}})\| \leq \alpha \big| X_{\text{test}}\big)$. The sequence $(f_k(X_{\text{test}}))_{k \in \mathbb{N}^*}$ converges to $\alpha$ in probability by (27), and is uniformly integrable since $|f_k(X_{\text{test}})| \leq 1$. Therefore, by the Lebesgue-Vitali theorem (Bogachev & Ruas, 2007, Theorem 4.5.4), $\lim_{k \to +\infty} \mathbb{E}[f_k(X_{\text{test}})] = \alpha$, which can be written as

$$\lim_{n_1 \to +\infty} \mathbb{P}\big(\|\mathbf{R}_k\big(S_{\text{test}}|X_{\text{test}}\big)\| \leq \alpha\big) = \alpha. \tag{28}$$

where we have used $k \to +\infty$ as $n_1 \to +\infty$. Hence, the sequence of random variables $(\|\mathbf{R}_k\big(S_{\text{test}}|X_{\text{test}}\big)\|)_{k \in \mathbb{N}^*}$ converges in distribution to a random variable drawn from the uniform distribution in $[0,1]$, $\text{Unif}([0,1])$.

In addition, $(\|\mathbf{R}_k(S_{n_1+i}|X_{n_1+i})\|)_{i=1}^{n_2}$ is a sequence of *i.i.d.* random variables, drawn from the same distribution as $\|\mathbf{R}_k(S_{\text{test}}|X_{\text{test}})\|$. This implies that the quantile function of $(\|\mathbf{R}_k(S_{n_1+i}|X_{n_1+i})\|)_{i=1}^{n_2}$ (denoted by $\hat{q}_n$) converges pointwise to the quantile function of $\|\mathbf{R}_k(S_{\text{test}}|X_{\text{test}})\|$ when $n_2 \to +\infty$ and $n_1$ is held fixed (this follows from the Glivenko-Cantelli theorem). Since we showed that $(\|\mathbf{R}_k(S_{\text{test}}|X_{\text{test}})\|)_{k \in \mathbb{N}^*}$ converges in distribution to a random variable distributed from $\text{Unif}([0,1])$, we conclude that $\hat{q}_n$ converges pointwise to the quantile function of $\text{Unif}([0,1])$ (denoted by $q$, which is continuous) as $n_1, n_2 \to +\infty$.

Moreover, in our setting where all distributions are bounded, $\hat{q}_n$ converges uniformly to $q$, see *e.g.,* Bogoya et al. (2016). In particular, since $\rho_{\lceil (n_2+1)\alpha \rceil} = \hat{q}_n\big(\lceil (n_2+1)\alpha \rceil / n_2\big)$, $\lim_{n_2 \to \infty} \lceil (n_2+1)\alpha \rceil / n_2 = \alpha$ and $q$ is continuous, one can show that $\lim_{n_1, n_2 \to +\infty} \rho_{\lceil (n_2+1)\alpha \rceil} = q(\alpha) = \alpha$.

$\square$

*Proof of Theorem 3.2.* Let $\mathbf{R}(\cdot|X_{\text{test}})$ be the OT rank map associated to the conditional distribution of $S$ given $X_{\text{test}}$. For a continuous distribution, the reference distribution of such OT rank map is associated to the random vector $R\Phi$ for two independent random variables $R$ and $\Phi$ drawn respectively from a uniform distribution on $[0,1]$ and on the unit sphere. Then,

$$\mathbb{P}\big(\|\mathbf{R}_k(S_{\text{test}}|X_{\text{test}})\| \leq \rho_{\lceil (n_2+1)\alpha \rceil}\big| X_{\text{test}}\big) - \alpha = \Delta_n + \Gamma_n, \tag{29}$$

where we have defined,

$$\Delta_n = \mathbb{P}\big(\|\mathbf{R}_k(S_{\text{test}}|X_{\text{test}})\| \leq \rho_{\lceil (n_2+1)\alpha \rceil}\big| X_{\text{test}}\big) - \mathbb{P}\big(\|\mathbf{R}(S_{\text{test}}|X_{\text{test}})\| \leq \rho_{\lceil (n_2+1)\alpha \rceil}\big| X_{\text{test}}\big),$$
$$\Gamma_n = \mathbb{P}\big(\|\mathbf{R}(S_{\text{test}}|X_{\text{test}})\| \leq \rho_{\lceil (n_2+1)\alpha \rceil}\big| X_{\text{test}}\big) - \alpha.$$

Both terms can be expressed in terms of cumulative distribution functions, *i.e.,*

$$\Delta_n = F_k\big(\rho_{\lceil (n_2+1)\alpha \rceil}|X_{\text{test}}\big) - F\big(\rho_{\lceil (n_2+1)\alpha \rceil}|X_{\text{test}}\big),$$
$$\Gamma_n = F\big(\rho_{\lceil (n_2+1)\alpha \rceil}|X_{\text{test}}\big) - \alpha,$$

where $F_k(\cdot|X_{\text{test}}), F(\cdot|X_{\text{test}})$ denote the c.d.f of $\|\mathbf{R}_k(S_{\text{test}}|X_{\text{test}})\|$ given $X_{\text{test}}$ and $\|\mathbf{R}(S_{\text{test}}|X_{\text{test}})\|$ given $X_{\text{test}}$ respectively. On the one hand, by the definition of conditional MK quantiles (del Barrio et al., 2024), $\|\mathbf{R}(S_{\text{test}}|X_{\text{test}})\|$ given $X_{\text{test}}$ is distributed from $\text{Unif}([0,1])$. Since $\lim_{n_1, n_2 \to +\infty} \rho_{\lceil (n_2+1)\alpha \rceil} = \alpha$ by Lemma B.2, and the c.d.f of $\text{Unif}([0,1])$ is continuous, then $\lim_{n \to +\infty} \Gamma_n = 0$. On the other hand, by eq. (27), $F_k(\tau|X_{\text{test}}) \xrightarrow{\mathbb{P}} F(\tau|X_{\text{test}})$ for every $\tau \in [0,1]$, which implies the uniform convergence of $F_k(\cdot|X_{\text{test}})$ to $F(\cdot|X_{\text{test}})$ by continuity of $F$, see *e.g.,* Eisenberg & Gan (1983). Therefore, $|\Delta_n| \leq \sup_\tau |F_k(\tau|X_{\text{test}}) - F(\tau|X_{\text{test}})| \xrightarrow{\mathbb{P}} 0$ as $n \to +\infty$. By (29), $\mathbb{P}\big(\|\mathbf{R}_k(S_{\text{test}}|X_{\text{test}})\| \leq \rho_{\lceil (n_2+1)\alpha \rceil}\big| X_{\text{test}}\big) \xrightarrow{\mathbb{P}} \alpha$ as $n_1, n_2 \to +\infty$, which concludes the proof.

$\square$

## C. Related works in multi-output regression

Multi-output conformal regression gathers increasing interest, as exemplified by several works (Braun et al., 2025; Luo & Zhou, 2025; Dheur et al., 2025; Klein et al., 2025) released concomitantly to the present paper. In particular, Klein et al. (2025) also motivate the use of transport-based quantiles to deal with multivariate scores. In this section, we provide an overview of possible methods for multivariate prediction regions and we refer the interested reader to Dheur et al. (2025) for a more complete comparison.

Methods based on component-wise univariate quantiles (Neeven & Smirnov, 2018) or ellipsoids (Messoudi et al., 2020; Johnstone & Cox, 2021; Henderson et al., 2024) were proposed as real-valued scores amenable to conformal multi-output regression. However, these methods could be phrased as multivariate center-outward quantiles in a generalized CP framework, hence the comparison in Section 3. Indeed, hyperrectangle regions based on componentwise quantiles and ranks have a long history when dealing with quantiles of multivariate data (Puri & Sen, 1971), although it implicitly assumes independent components. Under elliptic assumptions, Mahalanobis ranks are distribution-free and the associated quantiles provide an appropriate center-outward ordering (Hallin & Paindaveine, 2002). As a comparison, transport-based quantiles do not make any assumption on the shape of the underlying distributions. We refer to Hallin et al. (2021) for a recent review of existing notions of multivariate quantiles, with an emphasis on the distribution-freeness of ranks.

Hereafter, we point out other existing works to design real-valued scores while adapting to the underlying data structure. To ease the readibility, we follow the terminology proposed by Braun et al. (2025).

*Copula-based approaches.* Messoudi et al. (2021) introduced copulas in CP to model dependence in multivariate distributions. Copulas were also studied by Sun & Yu (2024) for time series datasets, and a main challenge addressed by Park et al. (2024) is their accurate estimation.

*Multiple testing approaches.* With a different perspective, Timans et al. (2025) propose a correction for CP phrased as multiple permutation testing. The reader might be interested in discussions and references therein for comparison with the classical Bonferroni correction. Additionally, one can note that multiple permutation tests were also proposed based on transport-based quantiles in Hlávka et al. (2025), although not dealing with CP.

*Latent-space approaches.* Recently, Feldman et al. (2023) combined multivariate quantiles and CP to obtain flexible prediction sets. Their method is based on a conditional variational auto-encoder for quantile regression, and they propose a calibration step to conformalize any quantile regression algorithms. In doing so, multivariate quantiles are used to accurately model the distribution of $Y|X$. By the way, note that OT-CP+ targets instead the conditional distribution of multivariate scores $s(X,Y)|X$, in order to cope with an arbitrary underlying algorithm $\hat{f}(x)$. We also mention a very recent work (Luo & Zhou, 2025) that transforms the distribution $Y|X$ to a Gaussian distribution based on Conditional Normalizing Flows, and combines this with volume minimization.

*Density-based and sampling-based approaches.* Other flexible approaches use density estimation (Izbicki et al., 2022), generative models (Wang et al., 2023) or both (Plassier et al., 2024) within the real-valued score function. We also mention the work of Sesia & Romano (2021) that incorporates conditional histograms. These approaches can provide discontinuous prediction regions, which is appropriate for multi-modal datasets.

*Optimization over template shapes.* Regions with convex shapes can be fitted to several clusters by volume minimization (Tumu et al., 2024), to induce multi-modal prediction regions. A similar optimization procedure has been recently proposed by Braun et al. (2025) to design the minimum-volume set that contains a proportion $\alpha$ of the data. Therein, the building blocks are arbitrary norm-based sets, with potentially multiple norms yielding non-convex prediction regions.

## D. Experimental Details

### D.1. Optimal transport solver

In all of our experiments, optimal transport problems are solved using the network simplex method implemented in the Python Optimal Transport library (Flamary et al., 2021). This solver is written in C++/Cython, making it generally faster than pure Python implementations.

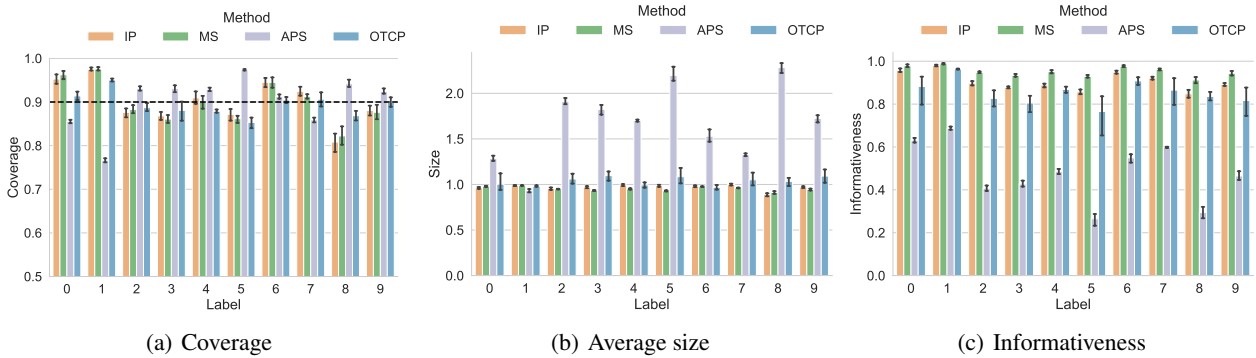

*Figure 10.* Label-wise results on $K = 10$ classes of MNIST

## D.2. Implementation details for regression

In Figure 2, the score $s(X, Y) = Y - \hat{f}(X)$ can be seen as a random vector $\zeta$ distributed as $\sum_{\ell=1}^{3} \pi_\ell \mathcal{N}(m_\ell, \Sigma_\ell)$, where $\pi_1 = \pi_2 = \frac{3}{8}, \pi_3 = \frac{1}{4}, m_1 = \binom{5}{0}, m_2 = -m_1, m_3 = \binom{0}{0}, \Sigma_1 = \begin{pmatrix} 4 & -3 \\ -3 & 4 \end{pmatrix}, \Sigma_2 = \begin{pmatrix} 4 & 3 \\ 3 & 4 \end{pmatrix}, \Sigma_3 = \begin{pmatrix} 3 & 0 \\ 0 & 1 \end{pmatrix}$.

For our real data experiments, we used datasets available in Tsoumakas et al. (2011). Table 1 specifies the number of observations and variables (for the features $X$ and for the output $Y$) for each dataset.

| Name | #Instances | #Features | #Targets |
|---|---:|---:|---:|
| atp1d | 337 | 411 | 6 |
| rf1 | 9125 | 64 | 8 |
| scm20d | 8966 | 61 | 16 |
| jura | 359 | 15 | 3 |
| wq | 1060 | 16 | 14 |
| enb | 768 | 8 | 2 |

*Table 1.* Details of datasets used for multiple-output regression

In the experiments involving OT-CP+ and a $k$-nearest neighbor search, setting $k = n/10$ for datasets of medium size and $k = \sqrt{n}$ for larger datasets provides a tradeoff between adaptive results and fast computational complexity.

## D.3. Implementation details for classification

The implementation of the ARS score relies on codes made available in the original paper (Romano et al., 2020b).

Experiments on MNIST and Fashion-MNIST in Figure 8 and Figure 9 involve a random forest classifier implemented with the Python library scikit-learn. We used 25 000 data splitted in train/calibration/test with ratio 10%/45%/45%, since this is sufficient for the classifier to reach 90% accuracy and to ensure reasonable size for the test data. Metrics are computed and averaged over $N = 10$ repeated random draws. Additional figures 10 and 11 related to these experiments are provided below, showing detailed label-wise results.

## D.4. Additional experiments

**Computational time.** In Figure 12, we report the runtimes of OT-CP and OT-CP+ for the empirical settings corresponding to Figures 2 and 3, with a varying number of calibration instances. As expected, OT-CP+ is slower than OT-CP, especially as the size of the calibration set increases.

**Experiments on real datasets.** Figure 13 complements experiments from Figure 4 with marginal coverage, volume and computational time. One can verify the appropriate marginal coverage of both methods, which legitimates the benefits of OT-CP+ in terms of adaptivity in Figure 4. One can observe that, for datasets with the lowest amount of samples, 'atp1d' and 'jura', OT-CP may tend to slightly overcover. Figure 13 also illustrates that OT-CP+ is more computationally intensive

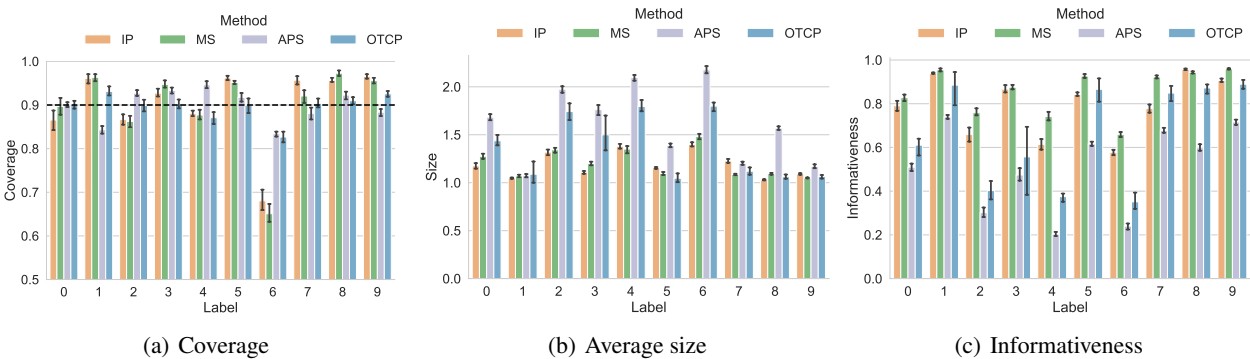

*Figure 11.* Label-wise results on $K = 10$ classes of Fashion-MNIST

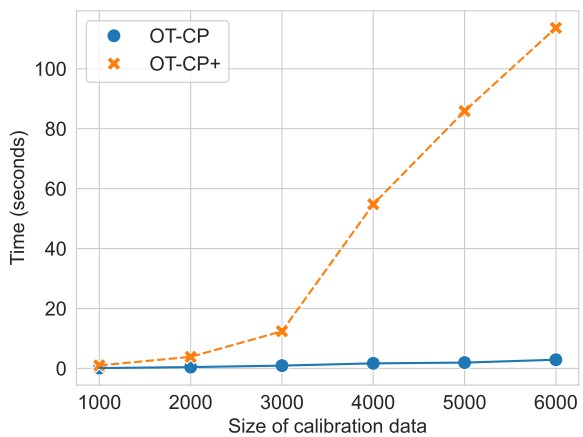

*Figure 12.* Computational time for OT-CP and OT-CP+ against the number of calibration instances

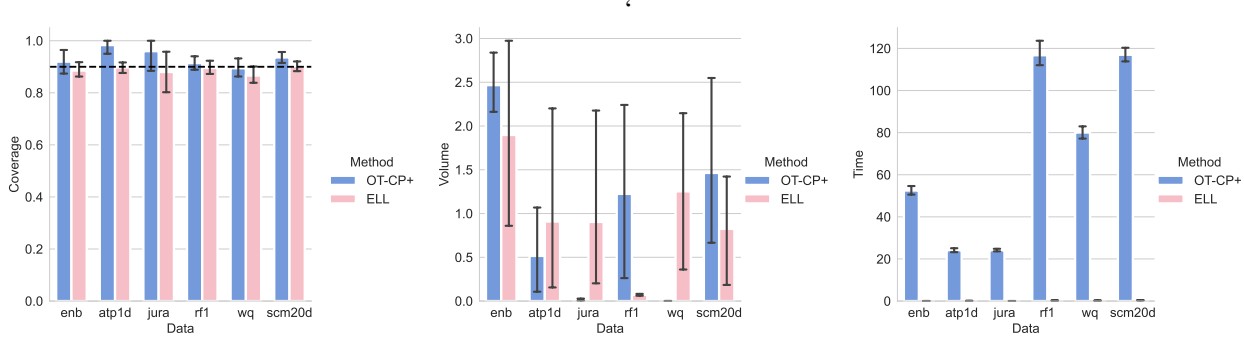

*Figure 13.* Marginal coverage, volume and calibration time (in seconds) for experiments of Figure 4

than computing covariances in the ellipsoidal approach `ELL`. Nevertheless, the experiments considered here take at most a few minutes, which remains reasonable.

**Additional results for classification.** Figure 7 illustrates OT-CP's ability to adapt to label confusions. Therein, classes 0 and 1 tend to be confused by the QDA classifier. In such cases, OT-CP achieves better trade-off between coverage and efficiency/informativeness. When the distribution is made easier (classes 0 and 1 become more distinct for QDA) all methods yield similar efficiency and informativeness. The latter setting is depicted in Figure 14. This supports that OT-CP can

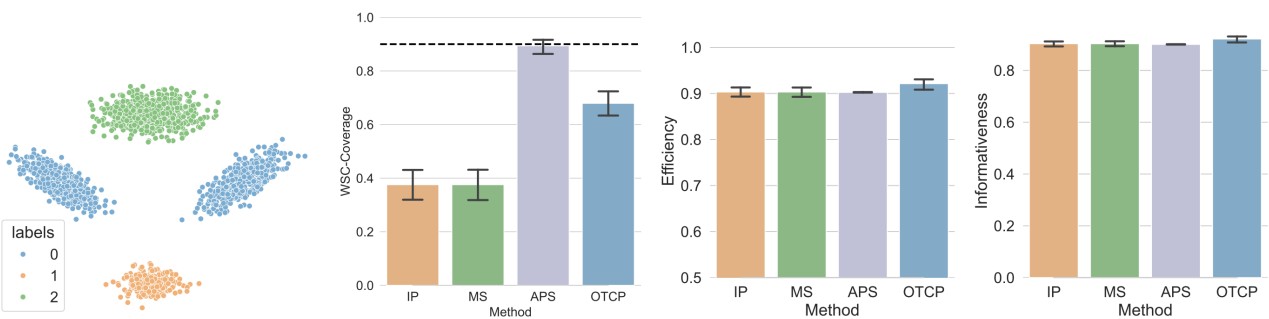

*Figure 14.* Classification by Quadratic Discriminant Analysis on simulated data, with a different uncertainty pattern

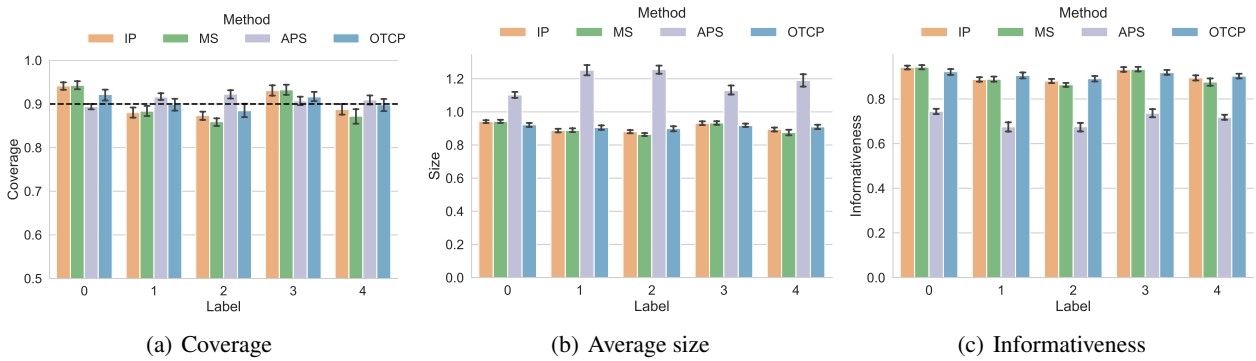

*Figure 15.* Label-wise results on $K = 5$ classes of MNIST

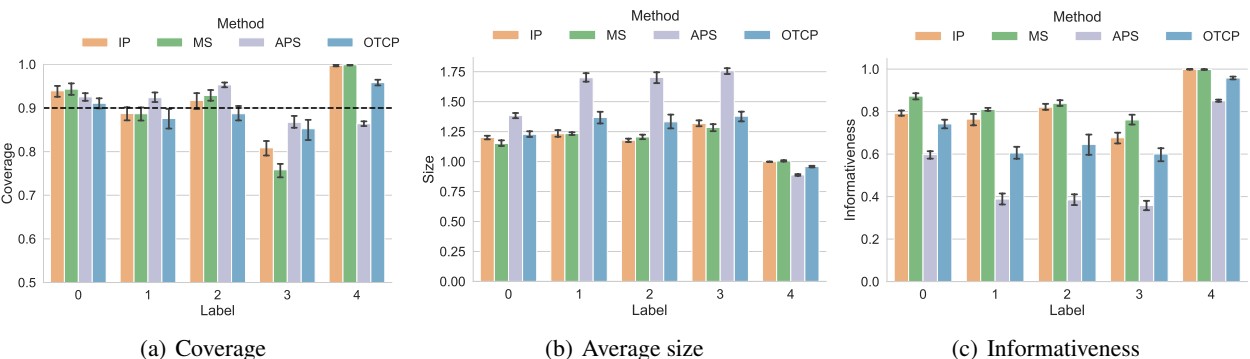

*Figure 16.* Label-wise results on $K = 5$ classes of Fashion-MNIST

effectively adapt to classification patterns where certain labels are prone to confusion.

In Figures 15 and 16, we conduct the same experiment as in Figure 9, but with a subset of $K = 5$ labels ($\{0, 2, 4, 6, 9\}$ for MNIST, and $\{$'T-shirt/top', 'Pullover', 'Coat', 'Shirt', 'Ankle boot'$\}$ for Fashion-MNIST). Similar conclusions as for Figure 9 apply here. Results in Figures 15 and 16 are averaged over 10 runs, each with 10 000 randomly chosen observations split in train/calibration/test with ratio $50\%, 40\%, 10\%$. We observe that OT-CP performs better for $K = 5$ than for $K = 10$, achieving the efficiency of the IP and MS scores while improving adaptivity, akin to the APS score. This suggests that for large $K$, OT-CP could benefit from more tailored score functions than $s(x, y) = |\bar{y} - \hat{\pi}(x)| \in \mathbb{R}_+^K$, inspired *e.g.,* by univariate penalized approaches (Angelopoulos et al., 2021; Melki et al., 2024).

