# OpenReview forum: "Optimal transport-based conformal prediction"
_ICML.cc/2025/Conference — ICML 2025 poster_

### Official Review · Reviewer_Gw2r · 2025-03-12

**Overall Recommendation:** 4

**Summary:**

The paper addresses multivariate conformal prediction and proposes an approach based on multivariate quantiles derived from optimal transport (OT), using a notion of multivariate statistical depth (MK depth) to define ranks. The application to both multivariate regression and classification is discussed, including a section on improved input-conditional coverage (with asymptotic guarantees). The approach is experimentally shown to improve conditional coverage while retaining useful set sizes against a few (simpler) baselines.

**Claims And Evidence:**

The key contribution is the use of OT-based multivariate quantiles which allows the handling of multivariate scores and does not pre-define prediction set shapes (e.g. rectangle = independence, or ellipsoids etc.) while incorporating data-dependent correlations. This results in perhaps less interpretable prediction set shapes but also makes it more data-driven, in line with CP goals. In general, the claims are backed up and convincing, albeit I remain a bit unconvinced of the practical usefulness of the method (see below) since shown experimental results and metrics seem carefully selected to support their method.

**Essential References Not Discussed:**

I've provided some suggested references above.

**Experimental Designs Or Analyses:**

I do not argue against the novelty of the proposed approach. My main gripe with this paper is the slightly insufficient experimental comparison and critical analysis of the method against related works, of which there exist many. For starters, the paper is lacking *both* a related works section and a discussion of limitations, namely practicality and computational complexity (it is only briefly mentioned in Remark 2.3).

To state a few more explicit experimental points:
- Fig. 2 seems a bit of an unfair comparison in the sense that OT-CP (their method) effectively accounts for data correlations, whereas the considered baselines are corrected with Bonferroni, subsuming an independence assumption across test dimensions. Given this fact, it is actually a bit surprising to see that their approach only fairs a bit better in terms of prediction set size. It would seem more fair to compare to other recent proposals exploiting correlations, e.g. [1,2].
- In Fig. 3b coverage within each partition / bucket is still marginal, right? And I would assume that as the buckets (I think of this as partition-conditional or mondrian CP) shrink, the conditional coverage would worsen as it approaches sample-conditional? So the claims on some sort of (exact) input-conditional coverage in finite samples do not hold true, e.g. as alluded in the caption of Fig. 3.
- Relatedly, sec. 3.2. with OT-CP would mean repeatedly solving the OT problem for every input sample based on its k neighbours. I am missing a proper discussion on practicality and runtime costs here. Is this really a useful approach?
- In Fig. 4 only worst-case coverage is shown, and the results seem to support the fact that OT-CP provides more balanced coverage. This is nice to have, but the considered baseline does not actually promise this form of coverage, right? Could we also see the marginal coverage results and prediction set sizes for the experiment in Fig. 4? Since if OT-CP tends to strongly overcover or provide overly large set sizes, then the worst-slab coverage results are not surprising.
- Since the empirical benefits (also for regression tasks) seem mainly in the realm of improving empirical conditional coverage (across various partitions), I would expect some comparisons to the multitude of recent CP methods addressing such tasks of conditional coverage, e.g. [4,5,6].
- Overall, there is substantially more recent work for multivariate CP that is entirely missed or omitted from this paper. I strongly encourage to have a look at e.g. [3] to see how a more thorough empirical evaluation could look like. I am not suggesting to implement the same breadth of comparisons, but it seems reasonable to consider or at the very least discuss a few more recent proposals. And again, in particular a meaningful discussion and reporting of empirical computational costs and runtimes seems necessary.

[1] Messoudi, Soundouss, Sébastien Destercke, and Sylvain Rousseau. "Copula-based conformal prediction for multi-target regression." Pattern Recognition 120 (2021): 108101.

[2] Timans, Alexander, et al. “Max-Rank: Efficient Multiple Testing for Conformal Prediction.” AISTATS (2025).

[3] Dheur, Victor, et al. "Multi-Output Conformal Regression: A Unified Comparative Study with New Conformity Scores." arXiv preprint arXiv:2501.10533 (2025).

[4] Romano, Yaniv, et al. "With malice toward none: Assessing uncertainty via equalized coverage." Harvard Data Science Review 2.2 (2020): 4.

[5] Sesia, Matteo, and Yaniv Romano. "Conformal prediction using conditional histograms." Advances in Neural Information Processing Systems 34 (2021): 6304-6315.

[6] Gibbs, Isaac, John J. Cherian, and Emmanuel J. Candès. "Conformal prediction with conditional guarantees." Journal of the Royal Statistical Society Series B: Statistical Methodology (2025):

**Methods And Evaluation Criteria:**

Experiments are provided for both regression and classification, against (only very few) relevant baselines. Results are compared in terms of common CP metrics which include (worst-case) coverage, set size, and fraction of singleton sets. Given the requirements to solve an OT problem for $n$ points (potentially multiple times if doing input-conditional) I would wish to also see more discussion and experimental results on computational costs and runtimes.

**Other Comments Or Suggestions:**

See above.

**Other Strengths And Weaknesses:**

- In Example 2 the motivation to use multivariate scores includes "This can be more helpful to capture the underlying confusion patterns of the predictor across different label modalities.". This motivation is never followed up on or shown in any way, so the exact benefit of working with multivariate scores versus collapsing them into a single dimension (but still accounting for correlations) remains a bit lacking
- How is the OT problem exactly solved in the experiments? It would be helpful to provide details on this (e.g. in the Appendix) to actually permit CP practitioners unfamiliar with OT to leverage this approach, beyond stating the OT problem only.
- Overall the paper reads quite well and is well structured, and I did not see any obvious typos. I appreciate that the authors provide several illustrative figures (e.g. Fig 1, 2a, 5, 6) that help visualize the concepts and make the intuitively understandable. The OT side of things is kept at a relatively high level which is fine, but could use a more thorough practical description in the appendix.

**Questions For Authors:**

Please see my comments and questions especially on the experimental design above.

**Relation To Broader Scientific Literature:**

There is no proper discussion of related works, so I believe the paper does not appropriately address the recent body of work (both on CP for conditional coverage and multivariate CP)

**Theoretical Claims:**

The use of OT-based multivariate quantiles is well-founded based on recent works in that direction, and well explained. The included Theorem 3.2 on asymptotic coverage under some assumptions seems reasonable, albeit I did not carefully check the proof. I appreciate that the regularity assumptions are clearly stated.

---

> ### Author Rebuttal · Authors · 2025-03-31
>
> We thank the reviewer for the careful reading and evaluation. We agree that our numerical experiments primarily compare against simple baselines. Our goal, however, is not to claim superiority over all methods but rather to motivate the use of transport-based quantiles in fundamental settings. We acknowledge that this may not have been sufficiently clear and we clarify this below. Thanks to your feedback, we also report additional metrics and results to provide a more comprehensive evaluation.
>
> *(computational costs and runtimes)*. We thank your for this comment which allows us to precise key properties of our methodology. Indeed, OT-CP+ requires solving multiple OT problems, which might be costly, especially for large-scale datasets. However, for the numerical experiments carried out in Fig.4, the computational time remains reasonable. An additional figure supporting this claim is now included: https://ibb.co/YTtv36ZM
>
> *(Comparison to existing related work)*. Our revised version now includes more related works as well as a discussion of limitations. Our main objective is to motivate the possibility of replacing univariate quantiles in the CP recipe by multivariate ones. OTCP should only be seen as a multivariate sorting of scores, which stands in contrast with the design of scores in applied settings. Our experiments suggest benefits of MK quantiles in terms of flexibility, but we do not claim to outperform all existing methods in multivariate regression. Rather, future work might profitably combine OTCP with more advanced multiple scores, different from componentwise residuals. Further numerical comparison is beyond the scope of our proposal and may perturbate the key message.
>
> *(Fig. 2)*. First, we stress that the ellipsoidal approaches do exploit correlations. Second, we intentionally only consider methods that can be interpreted as center-outward quantile regions, just as OT-CP, to ensure fair comparison. The mentioned references [1,2] differ in that they assume positive dependence, more related to a left-to-right ordering than a center-outward one.
>
> *(Fig 3b)*. We reported the empirical coverage averaged across subsets, which is drastically different from strategies that divide calibration examples into fixed groups (e.g., Mondrian CP). We also do not claim to achieve exact input-conditional coverage. We acknowledge the potential confusion and have revised our claim in Sec 3.2.
>
> *(Runtime cost of OT-CP+)*. Indeed, computational times is a limitation of OT-CP+, that is more costly than OT-CP. This cost depends on the parameter k. However, for experiments carried out in our paper, the price to pay seems relatively cheap, as illustrated in the new figure https://ibb.co/hpyCyFN comparing the time between OT-CP and OT-CP+ in the setting of Fig. 4.
>
> *(Type of coverage in Fig 4)*. The considered baseline shares our purpose of improving adaptivity, which is a common desiderata in CP. We agree that it does not guarantee asymptotic conditional coverage. In contrast, OT-CP+ satisfies this property at the price of larger set sizes on average when compared with local ellipsoids https://ibb.co/ymW5Mym6 A potential reason is that having more balanced conditional coverage requires including more points. We added a new figure tracking the marginal coverage of OT-CP+, to highlight that it does not tend to overcover https://ibb.co/1YNdSDny
>
> *(Recent CP methods for conditional coverage)*. As argued previously, our purpose is not to outperform all existing methods in conformal regression. This point has been clarified in the main body to ensure clarity when expressing our purposes.
>
> *(Discussion on related works)*. We discussed related work in "Introduction" and other paragraphs (CP methods for multi-output regression, in Sec. 3; CP methods for classification, in Sec. 4). We will integrate all these bibliographical elements into a dedicated subsection of the introduction.
>
> *Other Strengths And Weaknesses:* We appreciate your comments on the writing, structure and illustrative figures. We have added more details on how to solve OT in practice.
>
> *(Motivation for Example 2)* We believe that the results in Fig. 4 demonstrate OT-CP's ability to adapt to label confusions. Therein, classes 0 and 1 tend to be confused by the QDA classifier. In such cases, OT-CP achieves better trade-off between coverage and efficiency/informativeness. When the distribution is made easier (classes 0 and 1 become more distinct for QDA) all methods yield similar efficiency and informativeness: https://ibb.co/mVwMbsF6. This supports that OT-CP can effectively adapt to classification patterns where certain labels are prone to confusion.
>
> *(Solving the OT problem in practice)* We used POT library [R2], which solves OT with the simplex algorithm. This solver is written in C++/Cython and is thus generally faster than pure Python implementations. We have added these details in Appendix.
> [R2] Flamary et al. “POT: Python Optimal Transport.” JMLR 2021

---

> > ### Comment · Reviewer_Gw2r · 2025-04-05
> >
> > Thank you for the rebuttal and clarifications. Your answers satisfy most of my questions, but it is clearly visible in the additional figures (e.g. OT-CP+ vs. OT-CP; OT-CP vs. ELL) that the approach of combining CP with OT **does not**, at least in the presented form, convincingly outperform other recent proposals in the literature, and can tend to be both more computationally expensive and more conservative. Personally, this does not bother me as long as the limitations of the approach are **openly and clearly** stated, and the claims or contributions adjusted accordingly. I think there is sufficient novelty in the combination of these two fields to warrant acceptance, and there is no need to obfuscate details and try to promote the method beyond its abilities. Perhaps, as you suggested, future work can incorporate further ways to improve the conformal results.
> >
> > Such adjustments to the paper as well as dedicated related work and discussion sections would positively strengthen the paper in my opinion, and I hope the authors will take them to heart. **Overall I believe this paper warrants acceptance and is of interest to the CP community, and so I am raising my score.**

---

> > > ### Author Response · Authors · 2025-04-07
> > >
> > > Thank you for taking the time to read our rebuttal: we appreciate your recommendation to accept our paper. In line with your comments, we have included our additional results and discussion in our revised paper to provide a more nuanced and comprehensive evaluation of our method.
> > >
> > > In particular, we have added the following paragraph to our conclusion:
> > >
> > > "Approaching conformal prediction through the lens of optimal transport offers a unified framework for addressing a wide range of applications involving multivariate scoring functions, while maintaining rigorous theoretical coverage guarantees. In practice, these methods may be observed to be more conservative than existing score functions in the setting of multi-output regression. We identify opportunities for improvement in this direction by employing more suitable reference distributions for instance. Moreover, the flexibility of the approach, rooted in the Monge-Kantorovich quantile formulation, comes at the cost of increased computational complexity compared to conformal methods based on univariate scores. Building on these findings, future work will explore incorporating alternative transport-based methods into OT-CP to achieve better computational and statistical efficiency with respect to the scale of the problem."

---

### Official Review · Reviewer_bVvM · 2025-03-13

**Overall Recommendation:** 4

**Summary:**

The paper introduces OT-CP, a novel conformal prediction method for multi-output tasks based on optimal transport. The method constructs quantile regions for multivariate conformity scores while ensuring finite-sample coverage and achieving asymptotic conditional coverage. It introduces a new nonconformity score composed of two functions: a multivariate scoring function that preserves information about the error and a mapping function leveraging optimal transport to transform the multivariate score into a real-valued measure. This approach extends univariate conformal prediction by establishing a mapping between the predicted distribution and the uniform distribution on the unit ball. Applications in regression and classification demonstrate improvements in conditional coverage and/or volume.

**Claims And Evidence:**

Yes.

**Essential References Not Discussed:**

[1], based on a generative model, is another generative model that could be mentioned.

[1] Wang et al. “Probabilistic Conformal Prediction Using Conditional Random Samples.” AISTATS 2023.

**Experimental Designs Or Analyses:**

I find the empirical results unconvincing. OT-CP does not appear to stand out significantly in Figures 8, 9, and 10. The coverage is not necessarily better; in some cases, the prediction regions are larger, and the informativeness often seems lower compared to the IP and MS methods.

In instances where OT-CP appears to perform better, additional analysis is needed. For example, in the *Experiments on real data* section for regression with OT-CP+, it would have been helpful to report the average size of the prediction sets and other metrics evaluating conditional coverage in comparison to ELL.

**Methods And Evaluation Criteria:**

The method is evaluated on relevant synthetic and benchmark datasets using appropriate metrics, including marginal coverage, average size, and worst slab coverage. Experiments on synthetic and real data demonstrate that OT-CP outperforms ELL in regression (in terms of conditional coverage) and effectively balances coverage and informativeness in classification compared to IP, MS, and APS. However, the region size is not reported in the regression setting.

**Other Comments Or Suggestions:**

1) In Assumption 3.1, two quantities are denoted by $ \lambda $. Where do these quantities originate from?

2) In Example 2, the authors illustrate that a univariate score may fail to distinguish between the vectors (0.6,0.4,0) and (0,0.4,0.6), whereas a multivariate score can. I agree with this argument; however, later in the method, these vectors are mapped onto the unit ball. Given this transformation, isn’t there a high likelihood that the ranking induced by this mapping would still assign very similar ranks to these two vectors?

3) Could the choice of the unit ball shape influence the final prediction regions? If so, what is the rationale behind selecting the $ L_1 $-norm ball over other possible choices?

**Other Strengths And Weaknesses:**

#### **Strengths:**
- Exploring quantile regions for multivariate scores is a natural and worthwhile research direction.
- The framework is general and supports any multivariate conformity scores, making it applicable to both regression (e.g., residual-based scores) and classification (e.g., inverse probability scores).
- Theoretical guarantees are provided for both marginal and asymptotic conditional coverage.

#### **Weaknesses:**
- The **OT-CP+** method does not satisfy finite-sample marginal coverage, which is a fundamental requirement in conformal prediction.
- While computational aspects are briefly discussed, it remains unclear how **OT-CP+** compares computationally to **[1], [2], and [3]**. If I am not mistaken, the method requires computing the optimal transport map for each test input \( x \), which has a high computational complexity ($O(n^3)$ or $O(n^2)$ for approximations**), making it computationally demanding.
- Several relevant baselines are missing, particularly **[1], [2], and [3]** in regression. Notably, **[2]** can handle multimodal distributions, a capability that has not been discussed in this paper.
- In regression, the volume of the quantile regions has not been compared, limiting the assessment of efficiency.
- The motivation for using multivariate conformity scores is not entirely clear. Existing methods such as **[1], [2], and [3]** can already generate multivariate prediction regions without requiring multivariate conformity scores.
- The lack of publicly available code hinders reproducibility.

#### **References:**
- **[1]** Feldman, Shai et al. *Calibrated Multiple-Output Quantile Regression with Representation Learning.* JMLR (2023).
- **[2]** Wang, Zhendong et al. *Probabilistic Conformal Prediction Using Conditional Random Samples.* AISTATS (2023).
- **[3]** Sun, Sophia et al. *Copula Conformal Prediction for Multi-Step Time Series Prediction.* ICLR (2024).

**Questions For Authors:**

Please sea above.

**Relation To Broader Scientific Literature:**

This paper lies at the intersection of optimal transport and conformal prediction, both of which are active research areas.

Recent notable works in optimal transport include the introduction of quantile regions by Hallin et al. (2021), where vectors are ordered based on optimal transport, and their extension to regression by del Barrio et al. (2024).

In conformal prediction, several new methods have been proposed. The paper appropriately references copula- and ellipsoid-based approaches. Additionally, it acknowledges the more flexible method introduced by Feldman et al. (2023), which is particularly relevant to the discussion.

**Theoretical Claims:**

I checked the proof of Theorem 2.4 and the proof of Theorem 3.2, both of which appear to be correct.

---

> ### Author Rebuttal · Authors · 2025-03-31
>
> We thank the reviewer for the positive evaluation and address their comments with additional discussion and empirical results to illustrate the relevance and computational properties of OT-CP.
>
> *(Region size in regression)*. In Fig. 2, we reported the volumes to demonstrate that OT-CP achieves the desired coverage while producing smaller prediction sets than ELL, RECT. In Fig. 4, we initially did not include volumes since ELL fails to attain the desired conditional coverage, unlike OT-CP+. Based on your suggestion, we now monitor the volumes: https://ibb.co/ymW5Mym6
> Overall, our results suggest that OT-CP(+) achieve coverage while ensuring efficiency, whereas other CP methods either fail to attain coverage or do so by producing unnecessarily large prediction sets. ELL generates smaller regions than OT-CP+, which may explain why it fails to achieve the desired coverage. We will add these results in our paper.
>
> *(OT-CP in Figures 8, 9, and 10)*.  OT-CP provides more balanced coverage across classes, despite not being specifically designed for this task, while non-adaptive scores IP/MS can have low coverage (e.g., label 6 in Fig. 8). This benefit comes with better efficiency/informativeness than APS. This justifies our claim that OT-CP strikes "a favorable balance across all the considered metrics". Our revised version includes results averaged over labels: https://ibb.co/1GV3bsdq , https://ibb.co/nqTH1VCq.
>
> *(Average size of the prediction sets and other metrics  with OT-CP+)*. Additional metrics complementing Fig. 4 will be included: https://ibb.co/ymW5Mym6 , https://ibb.co/YTtv36ZM
> Volumes of OT-CP+ are often larger than those from the ellipsoidal approach, but the smallest average set size is not necessarily the best as argued in Angelopoulos and Bates (2023). The motivation for OT-CP+ is precisely to enhance adaptivity with respect to the input.
>
> *([1] could be mentioned)*. This relevant reference was already included in the initial submission ("Conclusion and Perspectives").
>
> *(OT-CP+ and marginal coverage).* We agree that the core guarantee of CP is finite-sample coverage. However, achieving adaptivity (i.e., obtaining prediction intervals with length adapted to the uncertainty relative to the considered test point), which is also a desirable property, is unattainable in finite samples. To introduce adaptivity, a sound strategy is to leverage universal consistent estimators, such as $k$NN. While this improves flexibility, it offers only asymptotic coverage, providing a different trade-off.
>
> *(OT-CP+ vs. [1], [2], and [3])*. A general comparison with [1-3] is an interesting direction for future work, but we explain below why it is beyond our current scope.
>
> *(Other relevant baselines)*. [1,2] were already cited and discussed in our initial submission. We now cite [3] and note that this work proposes a CP method tailored for time series forecasting. Comparing our approach to theirs would thus require adapting OT-CP to handle time series, which is challenging since extensions of MK quantiles to such data have never been studied.
>
> *(Multivariate prediction regions of [1], [2], and [3])*. Thank you for the question, which helps us refine our conclusions:
> - We do not claim OT-CP outperforms all CP methods across every metric, but rather present it as a general methodology, valid in regression settings, improving simple baselines with basic multivariate scores via optimal transport.
> - The suggested references are certainly relevant but appear to be more complementary than concurrent: [1] does not apply to black-box models, unlike OT-CP+ ; [2,3] propose CP strategies for handling complex univariate scores or time series.
> - Combining OT-CP and [1-3] can foster interesting future work. For instance, one can replace the ball-shaped regions with conformalized radii from [3] by our MK quantile regions for greater flexibility.
>
>
> *(Available code)*. We provided the code in the supplementary material.
>
> *(Assumption 3.1).* It implies $p(\cdot | x)$ is bounded away from $0$ and $+\infty$ on any compact subset of its convex support, a common assumption for transport quantiles (e.g., del Barrio et al., 2024).
>
> *("isn’t there a high likelihood that the ranking induced by this mapping would still assign very similar ranks to these two vectors?")*. Our mapping does not necessarily lead to similar rankings for these vectors, precisely because we use optimal transport rather than simple normalization. Therefore, MK ranks are $d$-dimensional vectors that integrate the overall geometry of the underlying distributions and capture directional differences (and not just the magnitude).
>
> *(Choice of the reference distribution)*. This is a relevant question, as choosing a reference distribution is an open research problem (even beyond CP) and is largely guided by heuristics. We chose the $L_1$-norm ball because the $L_1$-norm of our score yields IP, a commonly used univariate score in CP for classification: see below eq.(11).

---

> > ### Comment · Reviewer_bVvM · 2025-04-06
> >
> > We thank the authors for their response. We have increased our score.

---

> > > ### Author Response · Authors · 2025-04-07
> > >
> > > Thank you for your time and for raising the score.

---

### Official Review · Reviewer_1SF7 · 2025-03-16

**Overall Recommendation:** 4

**Summary:**

In the paper Optimal Transport-based Conformal Prediction, the authors propose a new conformal prediction framework leveraging optimal transport to produce multivariate score functions. They prove that such a framework also achieves distribution-free coverage. They validate the method for multi-output regression and multiclass classification using synthetic and real datasets. The results show how the proposed method is able to achieve asymptotic conditional coverage and provide adaptive predictive sets.

**Claims And Evidence:**

Everything seems reasonable to me.

The most conflicting point I find is the use of the k-nearest neighbor to extend OT-CP to OT-CP +. I would expect that the fix of the standard OT-CP to achieve conditional coverage could be solved in a more fundamental way. That is, using the k-nearest neighbor approach seems more like an approximation than a natural solution to the problem. The dependence on the $k$ hyperparameter

**Essential References Not Discussed:**

Most references w.r.t conformal prediction are discussed. Maybe regarding optimal transport more works could have been included, e.g. how to make more efficient the optimal transport problem. I believe this might be pointed out by other reviewers with more background on this field.

**Experimental Designs Or Analyses:**

The real datasets (lines 272) for the multi-output regression problem are not really explained, just referred to the reference work. I would add more information about which kind of datasets these are. Otherwise, it’s very difficult to tell the kind of regression problem we are facing.

**Methods And Evaluation Criteria:**

It wasn’t very clear when OT-CP+ or base OT-CP was being used during the experiments. I would make that more explicit, or even add both in the comparison.

Regarding the evaluation criteria, I think results per class in the classification section are a bit difficult to read. Maybe averaging across classes for the coverage and size of prediction sets is a more informative metric to report for the readers. And the refer to the Appendix the more granular results for class-specific coverage.

**Other Comments Or Suggestions:**

- In Figure 2 a), I would change the color of the prediction and MK prediction set, maybe: two different colors, or two shades of blue. It’s a bit difficult to read the dashed line with that shade of blue over the grey points.
- I find Figure 8 a bit difficult to read. At a first glance, it seems all methods are comparable. Would it make sense to average result for all classes, instead of outputting results per class? Maybe this way it’s easier to see the improvement using OT-CP. If following this suggestion, I would probably add the results for the other benchmark datasets, rather than having them in Appendix B.
- The use of $k$ for the $k$-nearest neighbor and $K$ for the number of classes in the classification problem can lead to errors when reading. I would maybe change the number of classes to a different variable (e.g. $L$, $M$?).
- For the results for regression, are we using OT-CP+, or OT-CP? It’s not clear from the caption and legends in the figures (Figure 2,3,4).

**Other Strengths And Weaknesses:**

I think the use of optimal transport to construct the conformity scores, both for classification and regression, is quite interesting and novel. This way the conformity score is able to adapt to the data distribution as shown in Figure 2b).

Regarding some weaknesses, I missed a computational comparison between baselines and the OT-CP in some tables or Figures. As far as I know, optimal-transport algorithms can be computationally quite demanding, and I worry about how practical would be the application of this work compared to simpler solutions already proposed in the conformal prediction literature. How does including the k-nearest neighbor step affect the computational complexity? Is it negligible?

Related to the point above, I missed the comments about the choice of the $k$ parameter in the $k$-nearest neighbor. How is this selected? Given that this is what yields to (asymptotic) conditional coverage, I think this is rather important.

**Questions For Authors:**

- How computationally complex is it to apply OT-CP? How much time does it take compared to the baselines?
- Can OT-CP be applied on full conformal prediction?
- Is the asymptotic conditional coverage determined by the k-nearest neighbor step of the algorithm?
- In Theorem 3.2, you state that, assuming $k \to \inf$ as $n \to \inf$, then $k/n \to 0$. Maybe I’m missing something trivial, but why? We also assume $n$ goes faster to infinity?
   - I just checked that it’s mentioned in the Appendix A.3 - $k$ is a function on $n$. You mention that you omit this dependency for clarity - could you add some note on that on the main paper/referral to the Appendix? Thanks
- Why do you assume $K \gt 3$ in the classification setup?
- How would OT-CP behave for a classification problem where we assume a One-vs-all strategy, i.e. using binary-to-multiclass framework based on sigmoids rather than softmax? Would results and assumptions still hold? Does this align with your future lines of work (lines 429-434?
- Figure 4, what is ELL? _ELL_ not really mentioned in the text.

**Relation To Broader Scientific Literature:**

N/A

**Theoretical Claims:**

Without diving very deep into the proofs, all claims made sense to me.

---

> ### Author Rebuttal · Authors · 2025-03-31
>
> Thank you for the positive and detailed feedback.
>
> *(OT-CP+ and the use of k-nearest neighbor)*. We agree that OT-CP+ is not the only way to make OT-CP adaptive: our proposed methodology aims to demonstrate that conditional coverage can be achieved with only a slight modification of the generic OT-CP framework. In addition to being easy to implent, the added k-NN step allows us to leverage established results on the consistency of quantiles (del Barrio et al., 2024), which serve as the foundation for our Theorem 3.2. Therefore, OT-CP+ should be seen as evidence of OT-CP's inherent flexibility, showing its ability to incorporate refinements to achieve specific properties, such as adaptivity.
>
> *(OT-CP or OT-CP+)*. The results for base OT-CP and OT-CP+ are resp. presented in Sections 3.1 and 3.2: Figure 2 for OT-CP, and Figures 3 and 4 for OT-CP+. The figure titles now explicitly indicate the used method.
>
> *(Numerical results per class)*. We will add results marginalized over $Y$ to improve readability: https://ibb.co/1GV3bsdq ; https://ibb.co/nqTH1VCq ; https://ibb.co/p6jGSmXj
> See also our answer "OT-CP in Figures 8, 9, and 10" for reviewer bVvm.
>
> *(Real datasets)*. In addition to the reference, we included a table in Appendix B.1 summarizing the main statistics for each dataset. Since these are directly sourced from the literature rather than created in our work, we felt that no further details were necessary. That said, we are open to add any relevant information if needed.
>
> *(Reference not discussed)*. Thank you for the suggestion. We will expand Remark 2.3 by referencing Sections 3 and 4 of Peyré & Cuturi (2019) for an overview of solvers for OT.
>
> *(Computational comparison between baselines and the OT-CP)*. The calibration time corresponding to Fig. 4 (OT-CP+ compared with ELL) will be added in the appendix, https://ibb.co/YTtv36ZM
>
> *(Computational complexity)*. Including the k-NN increases the required complexity, as illustrated in https://ibb.co/hpyCyFN. For all the experiments carried out in this paper (up to $n=10\,000$ points), the computational time of OT-CP is fast thanks to the Python Optimal Transport library (which solves OT with a simplex implemented in C++). OT-CP+ is more demanding than OT-CP, as it solves one OT problem per new test point. The parameter $k$ drives the computational time, and our Theorem 3.2 only requires that it grows slower than $n$ (see next answers).
>
> *(Choice of k in k-NN)*. As the reviewer correctly pointed out, $k$ should be chosen carefully to ensure asymptotic conditional coverage. More precisely, our Theorem 3.2 shows that k should grow to $+\infty$ as $n \to +\infty$ at a slower rate than $n$ (this point is further clarified below, in "Questions for Authors"). In our experiments, setting $k = n/10$ provides a tradeoff between adaptive results and fast computational complexity. Thank you for your comment: we agree that this is an important aspect to highlight that has been added in our paper.
>
> *(OT-CP and full CP)*. OT-CP can be adapted to full CP and would consist in solving OT on the entire training dataset and every candidate $(X_{test}, y)$. Similar to full CP methods, this would improve statistical efficiency and reduce variability by avoiding data splitting, at a price of increased computational cost.
>
> *(Asymptotic conditional coverage)*. Indeed, Theorem 3.2 applies to the quantile region computed with the $k$NN step, under some mild assumptions on $k$ as a function of $n$. We provide more explanations on this point below.
>
> *(Assumption on k and n)*. The assumption is $k \to +\infty$ as $n \to +\infty$ **and** $\frac{k}{n} \to +\infty$. This means $k$ grows to $+\infty$ as $n \to +\infty$, and at a slower rate than $n$ (for instance, $k = \log(n)$). This assumption is taken from del Barrio et al. (2024) and ensures the consistency of the sequence of weight functions in their Theorem 3.3, which we use (through their ensuing Corollary 3.4) to prove our Theorem 3.2. Note that this is a classical assumption to ensure the universal consistency of $k$NN estimators (e.g., Corollary 19.1 in [R1]).
>
> *(Classification with K>=3)*. Our method can also be applied to binary classification ($K = 2$) but our score (11) would have an intrinsic dimensionality of 1, since the vector of estimated class probabilities belongs to the simplex. Because this setting is not the most relevant for us, we chose to focus on $K \geq 3$.
>
> *(One-vs-all strategy)*. OT-CP can be applied to one-vs-all classification by simply selecting an appropriate score function (i.e., based on K sigmoids). Our theoretical results remain valid in this setting. This is an interesting research direction that could further broaden the applicability of OT-CP.
>
> *(ELL)*. ELL refers to the ellipsoidal approach with adaptive covariance estimation from Messoudi et al. (2022): see l.292-297. We have made this point clearer in our revised paper.
>
> [R1] Biau, Devroye (2015). Lectures on the nearest neighbor method.

---

> > ### Comment · Reviewer_1SF7 · 2025-04-08
> >
> > I would like to thank the authors for addressing my concerns and the other reviewers'. As **Reviewer Gw2r**, I still think that the proposed method might be lacking more improvements, especially the computational complexity, considering that the method often exhibits the same performance as other baseline methods. However, I believe the paper is still interesting for the CP community and that's why I raise my score.

---

> > > ### Author Response · Authors · 2025-04-08
> > >
> > > Thank you for your response and for increasing your score, we are glad that we addressed your concerns. In our revised version, we have added more discussion on the advantages and limitations of our methodology, in particular regarding the computational complexity, as outlined in our response to Reviewer Gw2r.

---

### Official Review · Reviewer_RqtM · 2025-03-20

**Overall Recommendation:** 3

**Summary:**

The authors tackle the problem of conformal prediction in regression and classification settings when the target random variable is multivariate. To do so, they use the quantiles definition as in [1], that defines the quantile function of a r.v. $Y \in R^n$ as the euclidean optimal transport map between a uniform random variable $U \in R^n$ and and $Y$.

[1] Chernozhukov, V., Galichon, A., Hallin, M., and Henry, M. Monge–Kantorovich depth, quantiles, ranks and signs. The Annals of Statistics, 2017

## update after rebuttal
My novelty considerations have been addressed in the rebuttal and so I increased the recommendation to weak accept.

**Claims And Evidence:**

They claim to "introduce a novel general CP framework" for multivariate conformal prediction. However, since the concepts of multivariate quantiles, ranks and confidence sets are already defined in [1] and derivative work, it is not clear to me what the main contribution is.

**Essential References Not Discussed:**

May be worth noting/including in the benchmarks the paper [2], that also applies the multivariate quantile function defined in [1] to regression tasks.

[2] Fast Nonlinear Vector Quantile Regression, AA Rosenberg, S Vedula, Y Romano, AM Bronstein, ICLR 2023

**Experimental Designs Or Analyses:**

The experiment design seems sound as it is described in the paper and the code is provided. I did not check the code.

**Methods And Evaluation Criteria:**

Method and evaluation are sound, there is both synthetic and real data and the target dimension is reasonable.

**Other Comments Or Suggestions:**

No.

**Other Strengths And Weaknesses:**

The paper is not super clear and the figures need clearer descriptions.

**Questions For Authors:**

I would like to have the main contribution of the paper better clarified.

**Relation To Broader Scientific Literature:**

The paper follows the work of [1] and its derivative papers but it is not clear to me what the main novelty is.

**Theoretical Claims:**

I did not check.

---

> ### Author Rebuttal · Authors · 2025-03-31
>
> ## General comment
>
> We would like to thank all the reviewers for their time and feedback. We have revised the paper accordingly and provide detailed responses below.
>
> 1. We emphasize that integrating multivariate quantiles into the conformal prediction framework while ensuring theoretical coverage guarantees is non-trivial. The property that ranks follow a uniform distribution, under exchangeability, is crucial. In the literature of multivariate quantiles, optimal transport tools extend this critical property (see, e.g., Hallin et al. 2021). However, the stability arguments invoked in standard proofs of the quantile lemma (see, e.g., Tibshirani et al. 2019) do not directly apply here, as they rely on unresolved theoretical questions in optimal transport. To better highlight this subtle point, we clarified Step 2 of our OT-CP methodology and explicitly demonstrate how our approach provides theoretical coverage guarantees.
> 2. We have added the suggested references to improve the discussion on related work.
> 3. We systematically computed the volumes of the prediction regions, precised the running times of the method, and conducted new numerical experiments to better support our claims.
>
> We now provide detailed reponses below. While we have carefully considered all reviewer comments, we are sometimes unable to provide an exhaustive answer for every point raised due to the character limit.
>
> [Tibshirani et al.,2019] Tibshirani, Foygel Barber, Candès, Ramdas, Conformal prediction under covariate shift, Neurips 2019
>
> [Hallin et al.,2021] Hallin, Del Barrio,  Cuesta-Albertos, Carlos Matràn, Distribution and quantile functions, ranks and signs in dimension d: A measure transportation approach, Annals of Statistics, 2021
>
> ## Answer to Reviewer RqtM
>
> We thank the reviewer for their comments. We appreciate the opportunity to clarify our contribution.
>
> *"However, since the concepts of multivariate quantiles, ranks and confidence sets are already defined in [1] and derivative work, it is not clear to me what the main contribution is."*
>
> While optimal transport-based quantiles are not new, the novelty of our work lies in integrating them into an effective and flexible conformal prediction (CP) framework designed for multivariate scores. This aligns with the CP literature, where quantiles typically serve as building blocks for uncertainty quantification strategies, rather than primary contributions. More precisely, unlike [1], we address several nontrivial challenges specific to Monge-Kantorovich quantiles for CP, including establishing both nonasymptotic and asymptotic coverage guarantees, selecting reference rank vectors $\{U_i\}_{i=1}^n$ that are appropriate for multivariate scores, and comparing performance with existing CP methods.
>
> *"The paper follows the work of [1] and its derivative papers but it is not clear to me what the main novelty is."*
>
> Our paper goes beyond prior work on transport quantiles, including [1], by integrating them in conformal prediction, which had not been explored before. By combining these two lines of work, we make the contributions outlined above but also offer a novel perspective that could be of interest to both the machine learning and optimal transport communities.
>
> *"May be worth noting/including in the benchmarks the paper [2], that also applies the multivariate quantile function defined in [1] to regression tasks."*
>
> We thank the reviewer for the additional reference [2], which we have added to the introduction. Nevertheless, we emphasize that this paper leverages the Monge-Kantorovich quantiles from [1] to address quantile regression, and not conformal prediction.

---

> > ### Comment · Reviewer_RqtM · 2025-04-07
> >
> > Thank you for the clarification, I will increase my recommendation.

---

> > > ### Author Response · Authors · 2025-04-08
> > >
> > > Thank you: we appreciate that you took the time to consider our clarification and that you recommend acceptance.

---

### Decision · Program_Chairs · 2025-05-01

**Decision:**

Accept (poster)

**Comment:**

The paper presents an OT based method for CP. Multiple reviewers found this interesting and useful for the CP community at large. One key advantage is that the prediction sets could potentially be non-convex. Simulations illustrate the advantage of the methodology.

All the reviewers opined an acceptance and three suggested clear acceptance. It is requested that the authors to incorporate all the reviewer suggestions appropriately in the final draft.